# SEGMENTED OPERATIONS USING MATRIX MULTIPLICATIONS

## ABSTRACT

Specialized computational units that perform small matrix multiplications as primitive operations are typically present in modern AI accelerators. However, these Matrix Multiplication Units (MMUs) are often underutilized for many fundamental deep learning operations besides dense matrix multiplications. Coincidentally, the lack of a rigorous theoretical model of computation for such architectures obstructs algorithmic design. In this work, we propose MMV-RAM, a computational model which judiciously extends the Vector-RAM model with an additional MMU. We provide a detailed theoretical analysis and carefully balance the computational power between the matrix and vector units, guided by the circuit complexity lower bound that parity is not in AC[0]. Given MMV-RAM, we proceed to algorithm design, starting with two fundamental parallel operations: *segmented scan* and *sum*. By expressing them as compositions of elementary parallel primitives (e.g., seg. sum reduces to: scan, compress, and vector differentiation), we can exploit MMUs to perform *speculative* blocked computations, ultimately leading to *provable theoretical speed-ups* against vector-only approaches. These results extend to other ubiquitous AI kernels, including dense matrix product, and sparse matrix-vector product. As a case study, we implemented the proposed algorithms on the Ascend 910B AI accelerator, which contains matrix and vector cores. We evaluate these implementations on synthetic and real-world datasets from various applications, including Large Language Models.

## 1 INTRODUCTION

In recent years, the unexpectedly rapid evolution and development of Artificial Intelligence (AI) and, in particular, deep learning, have strongly influenced the computing industry. Deep learning workloads, both training and inference, demand more and more performance every year (Kaplan et al., 2020). The high demand for accelerated AI computing has resulted in manufacturing domain-specific accelerators (often referred as Neural Processing Units (NPUs)) that contain, most notably, specialized MMUs. Examples of such MMUs on massively-produced devices include NVIDIA Tensor Cores (NVIDIA Corporation, 2020), Google TPU Matrix Multiply Unit (Jouppi & et al., 2021), and Huawei Ascend Cube Unit (Liao et al., 2021), to name a few.

Although MMUs have proven highly effective to increase the peak arithmetic performance for regular and dense computations, such as convolutions, the attention mechanism in transformers (Vaswani et al., 2017), and fully connected layers in deep learning, their applicability to irregular and sparse computations remains uncertain. Indeed, sparsity in deep learning is a well-known characteristic that has not yet been successfully exploited in practice (Hoefler et al., 2021; Dave et al., 2021; Chen et al., 2022). This fact has raised a debate in the computing industry on whether specialized computing units for sparse computations, usually referred to as "Sparse Tensor Cores", should be designed (Wang et al., 2021; Mishra et al., 2021; He et al., 2020), *or* if irregular and sparse computations should be handled using the already existing MMUs along with algorithmic innovations (Chowdhury et al., 2020; 2021; Zachariadis et al., 2020; Okanovic et al., 2024; Kung et al., 2019).

In this work, we follow the latter approach, exploiting existing MMUs for algorithmic design, as initiated in (Chowdhury et al., 2021). The two main problems that we target are segmented operations, specifically, segmented sum and scan, two fundamental computational primitives in parallel computing with numerous applications (Blelloch, 1990a; Sengupta et al., 2007; Chatterjee et al., 1990).

Importantly, segmented operations have been proven highly effective for irregular and sparse computations, such as sparse matrix-vector multiplication (Blelloch et al., 1993; Liu & Vinter, 2015b), which is of great importance to efficiently address such computations in deep learning workloads.

The analysis of parallel algorithms, especially for prefix problems, is well-established in parallel models of computation such as the Parallel RAM (PRAM) (Shiloach & Vishkin, 1981; JáJá, 1992; Dhall, 1994), Bulk Synchronous Parallel (BSP) (Valiant, 1990; McColl, 1993; 2005; Tiskin, 1998), LogP (Culler et al., 1993), Vector-RAM (Pratt et al., 1974; Blelloch, 1990b), the TCU Model (Chowdhury et al., 2020; 2021), and circuit models (Dhall, 1994; Vollmer, 1999; Wegener, 1987; Ladner & Fischer, 1980). As we target AI accelerators, TCU and Vector-RAM are arguably the most relevant models, but none of them fully embraces the heterogeneity of modern accelerators.

The TCU model is the first to explicitly target accelerators with MMUs. However, it disregards Vector Units (VCUs). This omission is both technically non-trivial to address (as we thoroughly discuss in this work), and essential for modeling modern architectures. Indeed, VCUs are typically present near the MMUs in AI accelerators, since element-wise activation operations are ubiquitous in AI workloads. For example, in convolutional neural networks, convolutions are followed by the RELU activation function (Krizhevsky et al., 2012), and, in the transformer architecture, the linear layers of FFN are followed by the specialized GELU function (Vaswani et al., 2017).

The Vector-RAM model, instead, also presents some limitations in this context. First, it does not entail an explicit representation of MMUs, making it unbefitting for analyzing algorithms that use such operations. Moreover, its vector instruction set is overly powerful: operations such prefix sums, or even matrix multiplication, can be performed in a single (parallel) step. With such a powerful vector unit, an explicit matrix multiplication unit becomes obsolete.

## 1.1 CONTRIBUTIONS

To address the aforementioned limitations of existing models, we propose an extension of the Vector-RAM model, called *MMV-RAM*, which includes an additional $\underline{M}$atrix $\underline{M}$ultiplication unit, along with the $\underline{V}$ector unit. The MMU handles matrix products of sizes $n \times s$ and $s \times s$ in a single step ($s \geq 2$ is a model parameter and $n$ is arbitrary), whereas the VCU is used for elementary vector operations.

Our key technical contribution in the design of MMV-RAM is the separation and balancing of computational power between the matrix and vector units, ensuring that they complement rather than subsume each other. This balance (formally proved in Theorem 3.1) is grounded in a fundamental result from circuit complexity: the parity function cannot be computed by constant-depth, polynomial-size (denoted by AC[0]) circuits (Furst et al., 1984; Yao, 1985; Håstad, 1986). Therefore, as long as the vector unit of MMV-RAM is restricted to AC[0], then it cannot efficiently simulate the matrix unit, thus maintaining a clear computational distinction between the two units.

MMV-RAM adopts a standard work/depth cost framework, enabling an intuitive analysis of algorithms. For algorithm engineers, the analysis becomes straightforward in MMV-RAM: the number of matrix and vector operations gives the total number of steps of the parallel algorithm ("time/step complexity"), while the total "cost" or "work" is the sum of their corresponding costs.

Given the definition of MMV-RAM, we next design and analyze algorithms, starting with segmented operations. We first observe that these operations can be written as compositions of elementary parallel primitives, which lead to our main Algorithms 4.1 and 4.2. Our analysis indicates that by exploiting the MMU of MMV-RAM we can obtain *provable theoretical speed-ups* against vector-only implementations. In fact, we are able to an *explicit trade-off* between the MMU size parameter $s$, and the achievable speed-ups. Our main results are summarized in the following informal theorem.

**Theorem 1.1** (Informal, see Thms. 4.2 and 4.3). *In MMV-RAM, there exist algorithms for segmented sum and scan of length $n$, which require $\mathcal{O}(\log_s(n))$ steps, and $\mathcal{O}\left(nB(s + \frac{B}{s})\right)$ work, where $B$ is the number of bits-per-element. In contrast, any algorithm that uses only vector operations and executes work that scales polynomially in $n$ requires $\Omega\left(\frac{\log(n)}{\log(\log(n))}\right)$ steps.*

As direct applications, we extend these results for higher level algorithms, using segmented operations as subroutines (detailed in Table 4.1). The first example is the element-wise multiplication of two vectors with integer elements. It is well-known that integer multiplication is not in AC[0], since parity reduces to it. As such, this operation cannot be directly implemented on the VCU in constant

depth. But it can be efficiently implemented using both the MMU and VCU using segmented operations. Ultimately, we also discuss algorithms for the product of two dense matrices, and well as the product between a sparse matrix and a dense vector (SpMV). For all these applications we report theoretical speed-ups similar to those of Theorem 1.1. Our contributions are summarized as follows:

1. We propose the MMV-RAM model of computation, as an extension of Vector-RAM, that contains both matrix and vector processing units, targeting modern AI accelerators.

2. In MMV-RAM, we design and analyze algorithms for segmented operations (Algs. 4.1, 4.2) and their applications to other kernels, including matrix multiplication and SpMV. We prove that by exploiting the MMU to perform speculative blocked operations, we can achieve theoretical speed-ups compared to *any* vector-only MMV-RAM algorithm.

3. We experimentally evaluate the proposed algorithms on the Ascend 910B AI accelerator, which features both MMUs and VCUs, on real-world and synthetic datasets. By effectively utilizing the MMUs, we report substantial performance gains for segmented scan compared to a vector-only baseline. For sparse-attention datasets, we also report noteworthy speed-ups for SpMV compared to highly optimized, multi-threaded CPU libraries.

**Notation.** Matrices and vectors are denoted by capital and small letters, respectively, in bold font. All logarithms are base two, unless explicitly specified. For a matrix $\mathbf{A}$, $\|\mathbf{A}\|_{\max}$ is the maximum absolute value over all matrix elements. We denote by $\mathbf{U}_s$ the upper-triangular all-ones square matrix of size $s$. All vectors are considered column vectors, and $\mathbf{A}^\top$ and $\mathbf{x}^\top$ denote the transpose of matrix $\mathbf{A}$ and vector $\mathbf{x}$, respectively. Vectors and matrices are zero-index based. For a vector $\mathbf{x}$, $\mathbf{x}(i)$ is the $(i+1)$-th entry, and for integers $i < j$, we denote by $\mathbf{x}(i:j)$ the subvector $(\mathbf{x}(i), \mathbf{x}(i+1), \ldots, \mathbf{x}(j-1))$. $\mathbf{x}(i:j:s)$ is the $s$-strided subvector with offset $i$: $(\mathbf{x}(i), \mathbf{x}(i+s), \mathbf{x}(i+2s) \ldots, \mathbf{x}(j-1))$. If $j$ is omitted, that is, when we write $\mathbf{x}(i::s)$, then it defaults to the length of $\mathbf{x}$. For a vector $\mathbf{x}$ of length $n$, the operation $\mathbf{z} \leftarrow \mathsf{MATMUL}(\mathbf{x}, \mathbf{A})$ denotes that $\mathbf{x}$ is viewed as a matrix $\mathbf{X}$ of size $\lceil n/s \rceil \times s$, in a row-major order (zero-padded if necessary), and the result $\mathbf{z}$ is a vector containing the elements of the matrix product $\mathbf{XA}$, in row-major order as well. "Hat" accents on vectors, e.g. $\widehat{\mathbf{x}}$, denote the scanned version of the vectors. The "bar" accent typically denotes that all consecutive blocks of length $s$ are prefix-summed within each block, e.g., $\overline{\mathbf{x}} \leftarrow \mathsf{MATMUL}(\mathbf{x}, \mathbf{U}_s)$.

**Outline.** In Section 2, we recall some basic definitions and background. Section 3 contains the analysis of the MMV-RAM model. The main algorithms for segmented operations and their applications are described in Section 4. In Section 5, we provide an experimental evaluation of the proposed algorithms, before concluding in Section 6.

## 2 BACKGROUND AND DEFINITIONS

In this section, we define the segmented scan and segmented sum tasks. In the (unsegmented) scan task, we are given an array $\mathbf{x}$ of length $n$ and the task is to return an array $\mathbf{z}$ of length $n$ such that $\mathbf{z}(i) = \sum_{j \leq i} \mathbf{x}(j)$ for each $i$. Segmented scan is a generalization of the scan task, defined as follows:

**Problem 2.1** (*Segmented Scan*)**.** *Given two arrays $\mathbf{x}$ and $\mathbf{f}$ both of length $n$, where $\mathbf{f}(i)$ is a boolean value indicating whether $\mathbf{x}(i)$ is the first element of a segment, the segmented scan returns an array $\mathbf{z} = (\mathbf{z}(0), \mathbf{z}(1), \ldots, \mathbf{z}(n-1))$ such that:*

$$\mathbf{z}(i) = \begin{cases} \mathbf{x}(i), & \text{if } i = 0 \text{ or } \mathbf{f}(i) = 1, \\ \mathbf{z}(i-1) + \mathbf{x}(i), & \text{otherwise.} \end{cases} \tag{1}$$

*Segmented sum* is the collection of the last elements of each segment of the segmented scanned array. As an example, assume $\mathbf{x} = (2, 2, 3, 3, 1, 3, 1, 2)$ and $\mathbf{f} = (1, 0, 1, 0, 0, 1, 0, 0)$. The segmented scan and sum are $(2, 4, 3, 6, 7, 3, 4, 6)$ and $(4, 7, 6)$, respectively. For our analysis, we also define the COMPRESS that returns all entries $\mathbf{x}(i)$ where the corresponding $\mathbf{f}(i) = 1$. For the example above, COMPRESS$(\mathbf{x}, \mathbf{f})$ returns $(2, 3, 3)$, while COMPRESS$(\mathbf{z}, \mathbf{f}^-)$ is the segmented sum of $\mathbf{x}$ over $\mathbf{f}$, where $\mathbf{f}^- = (\mathbf{f}(1:), 1)$.

In the literature, scan problems typically assume a binary associative operator, i.e., a semi-group structure (see, e.g., (Blelloch, 1990a; Dhall, 1994)). Here, our algorithms require that the set of elements and the scan operator must form a group.

**Circuits.** In our analysis we use results from circuit complexity theory. A circuit is a Directed Acyclic Graph (DAG) $G = (V, E)$. Nodes with zero in-degree are called *inputs*, and nodes with zero out-degree are called *outputs*. $|E|$ is the *size* of the circuit, while its *depth* is the longest path from an input to an output. The *fan-in* is the largest in-degree, and the *fan-out* is the largest out-degree. The vertices are also called *gates*, and they perform mathematical operations on their predecessors.

In this work we consider *boolean circuits*, which consist of $\{\wedge, \vee, \neg\}$ gates that operate on boolean variables. The $\wedge$ and $\vee$ gates have unbounded fan-in by default, unless specified otherwise. For $i \in \mathbb{N}$, NC[i] and AC[i] are the classes of boolean circuits with fan-in two and unbounded fan-in, respectively, having $n^{O(1)}$ size and $\mathcal{O}(\log^i(n))$ depth, for input length $n$. It is known that NC[0]$\subset$ AC[0] $\subset$ NC[1] (Vollmer, 1999, Corollary 4.35). We consider *uniform families of circuits*, i.e., their description does not change significantly for different input sizes (Borodin, 1977; Vollmer, 1999).

## 3 MMV-RAM: A MODEL FOR MATRIX MULTIPLICATION AI ACCELERATORS

In this section we define the MMV-RAM model, by judiciously extending the Vector-RAM model with an MMU. As shown in Figure 3.1, MMV-RAM contains the following three processing units:

- A scalar unit, which performs arithmetic and logic operations, and address calculations;
- A VCU, which performs a single vector operation in every step;
- An MMU, that multiplies two matrices of size $n \times s$ and $s \times s$ in a single step.

The positive integer $s$ is the only parameter of the model, which corresponds to the (right) matrix multiplication size and it is independent of the input length[1]. To be more precise, the MMU is a uniform family of circuits (Borodin, 1977; Vollmer, 1999) that is assumed to perform a single-step matrix multiplication. The VCU in MMV-RAM implements a fixed set of instructions defined by a uniform family of AC[0] circuits, i.e., boolean circuits with unbounded fan-in, constant depth, and polynomial size. Table 3.1 summarizes how existing AI accelerators map on the proposed MMV-RAM model. In Appendix A.1 we discuss further extensions and variants of MMV-RAM.

Table 3.1: Examples of architectures where MMV-RAM can be used for algorithmic analysis.

| Architecture | Matrix Mult. Unit | Vector-like Unit |
|---|---|---|
| NPU (AMD) (Hunhoff et al., 2025) | Systolic array | Compute Tile |
| NPU (Huawei) (Liao et al., 2019) | Cube Unit | Vector Unit |
| NPU (Tesla) (Talpes et al., 2020) | MAC Core | SIMD Unit |
| GPU (AMD) (AMD, Inc., 2025) | CDNA Matrix Core | SIMD Unit |
| GPU (NVIDIA) (Markidis et al., 2018) | Tensor Cores | CUDA Cores |
| TPU (Google) (Jouppi & et al., 2017) | Matrix Multiply Unit | Vector Proc. Unit |

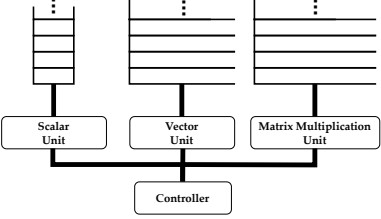

Figure 3.1: MMV-RAM model.

**Work/depth cost model.** An MMV-RAM algorithm is evaluated using *work* and *depth* analysis as in the Vector-RAM model (Blelloch, 1996; 1990b). At each *step*, the machine can execute a scalar instruction, a vector instruction, or a matrix multiplication. To keep the cost model of MMV-RAM simple, we focus on bounding the number of VCU and MMU calls (i.e. *step* complexity), and their length, which implies the *work* complexity. The latter represents the total number of basic operations that are executed in each step, potentially in parallel. More formally:

- $\mathcal{M}(n)$ is the work for matrix multiplications between two $n \times s$ and $s \times s$ matrices.
- $\mathcal{V}(n)$ is the work for vector instructions on vectors of length $n$.

$\mathcal{M}(n)$ depends on the number of rows of the (left) input matrix, and the bit-width of the matrix elements. It scales linearly in $n$, which means that we can write $\mathcal{M}(an) = a\mathcal{M}(n)$ and $\mathcal{M}(n + a) = \mathcal{M}(n) + \mathcal{M}(a)$. Note that we do not assume a specific circuit implementation for the MMU, but rather define it only based on the aforementioned properties. For example, a constant-depth threshold circuit (TC[0]) that implements the standard dot-product based algorithm satisfies these properties. We refer to (Parekh et al., 2018) and Appendix A.2 for details on matrix product circuits.

---

[1]Typical values of $s$ in existing architectures are $16, 32, 128$ or $256$.

$\mathcal{V}(n)$ depends on the length of the input vector and on the number of edges (size) of the corresponding AC[0] circuit that implements it. To be precise, VCU instructions are uniform families of unbounded fan-in circuits with constant depth and polynomial size in *all the involved parameters*, i.e., the input length $n$, the number of "bits-per-element" $B$, and the model parameter $s$. The only instructions that are assumed "by default" in the model are element-wise $\wedge, \vee$, and $\neg$. If complex vector instructions are used by an algorithm, their associated circuit sizes, depths, fan-in, and fan-out must be explicitly defined. Some examples can be found in Appendix A.3.2.

**Memory accesses.** The VCU can read $n$ consecutive words from the main memory in one step, where $n$ is arbitrary. The MMU can also read $ns$ consecutive words from the main memory in a single step. These words are either stored in the $n \times s$ or in the $s \times s$ buffer of the unit. The input and output matrices in MMV-RAM can have any *fixed* matrix layout, say row or column major. Without loss of generality, we assume row-major order. Matrix transposition and irregular/strided memory accesses are *not* explicitly allowed, but they can be implemented with a dedicated AC[0] circuit family in the VCU. In this case, a precise circuit definition is necessary.

**Remark 3.1.** *Arithmetic operations can increase the number of bits required to store the (intermediate) results. For example, integer multiplication can double the number of bits. For each algorithm we upper bound the number of required bits to avoid overflow, since it affects the total complexity.*

### 3.1 THE BALANCE BETWEEN THE MATRIX AND VECTOR UNITS

We now argue that both units of MMV-RAM are necessary to design efficient algorithms: the MMU cannot be efficiently simulated by the VCU, but the VCU provides crucial capabilities that are hard (or even impossible) for the MMU. The foundational result that our analysis builds upon is the circuit complexity lower bound that parity is not in AC[0] Furst et al. (1984); Yao (1985); Håstad (1986). By exploiting this result, and with some additional arguments, we can prove the following theorem.

**Theorem 3.1.** *In MMV-RAM, the MMU and the VCU complement each other in the following sense:*

- *The VCU can compute non-linear element-wise functions, such as RELU, on a vector $\mathbf{x}$ in a single step, while the MMU cannot execute such operations.*

- *The MMU can multiply two matrices $\mathbf{A}$ and $\mathbf{B}$ with sizes $n \times s$ and $s \times s$, in one step. However, any MMV-RAM algorithm that uses only the scalar and vector units to compute $\mathbf{AB}$, and executes work that scales polynomially in $s$, requires $\Omega\left(\frac{\log(s)}{\log(\log(s))}\right)$ steps.*

- *There exists a linear vector transformation (a permutation) which can be efficiently implemented in the VCU in one step and linear work, but it requires $\Omega(s)$ calls to the MMU.*

*Proof.* The full proof can be found in Appendix A.3.1. In brief, the MMU cannot execute RELU because it can only perform linear transformations. The second item relies on the size-depth trade-off by Håstad (1986) for the parity problem, while the third item is detailed in Lemma A.1. $\square$

## 4 SEGMENTED OPERATIONS IN MMV-RAM

Given the definition of MMV-RAM, we next proceed to algorithmic design. Our main algorithms rely on unsegmented scan as a parallel primitive to perform speculative computations. For this, we use the unsegmented scan algorithm of (Zouzias & McColl, 2023), which is an unbounded fan-in generalization of the classic algorithm of (Brent & Kung, 1980). Theorem 4.1 reports its complexity in the MMV-RAM model. The work-depth bounds match the ones of (Zouzias & McColl, 2023) with respect to $n$ and $s$, but additionally here the MMV-RAM model explicitly takes into account the gather/scatter vector operations. The proof of Theorem 4.1 is detailed in Appendix A.3.3.

**Theorem 4.1** (SCAN). *Given a vector $\mathbf{x}$ with integer entries bounded by $\mathbf{x}_i \leq M$, for some $M$, we can compute a vector $\mathbf{z}$ that contains the scan of $\mathbf{x}$ in the MMV-RAM model using the algorithm of (Zouzias & McColl, 2023) (listed in Algorithm A.1). The algorithm requires:*

$$\mathcal{O}(\log_s(n)) \text{ steps}, \quad \mathcal{O}\left(\mathcal{M}(\tfrac{n}{s}) + nB^2\right) \text{ work},$$

*and $B \in \mathcal{O}(\log(nM))$ bits-per-element to avoid overflow. On the contrary, any MMV-RAM algorithm that uses only the VCU and executes $n^{\mathcal{O}(1)}$ work requires $\Omega\left(\frac{\log(n)}{\log(\log(n))}\right)$ steps.*

## 4.1 SEGMENTED SUM (SCD) AND SEGMENTED SCAN (SSCR)

Having SCAN as a building block, we next analyze our main algorithms for segmented operations. First, we present a segmented sum algorithm, called SCD (Scan, Compress, and Differentiation, Algorithm 4.1). In the first step of SCD, the unsegmented scan of the input $\mathbf{x}$ is computed speculatively. In the second step, $\mathbf{f}$ is slightly modified to $\mathbf{f}^-$ so that the end positions of all segments are marked by the ones in $\mathbf{f}^-$. Indeed, $\mathbf{f}^-$ is obtained by $\mathbf{f}$ by shifting the vector to the left and appending 1 at the end. Then, the scanned values of $\mathbf{x}$ corresponding to the end position of all segments are collected through a parallel compaction / compress operation using $\mathbf{f}^-$. One can verify that the segmented sum can be obtained by applying vector differentiation $\mathbf{z}_{\text{diff}}(i) = \mathbf{z}(i) - \mathbf{z}(i-1)$ (assume $\mathbf{z}(-1) = 0$) on the output vector of compress.

Now, we argue that all three steps of SCD could benefit from a MMU, i.e., SCD is amenable to an efficient implementation in the MMV-RAM model. SCAN can be implemented using matrix multiplications as shown in Theorem 4.1. Similarly, parallel COMPRESS is can be implemented by first scanning the input flag vector to calculate the output indices. Finally, the DIFF step of SCD can also employ the MMU as described in Appendix A.3.4 (albeit with low utilization).

| **Algorithm 4.1** Segmented Sum $\cong$ DIFF ∘ COMPRESS ∘ SCAN | **Algorithm 4.2** Segmented Scan $\cong$ REVERT ∘ COMPRESS ∘ (SCAN, SCAN) |
|---|---|
| 1: **procedure** SCD($\mathbf{x}, \mathbf{f}$) | 1: **procedure** SSCR($\mathbf{x}, \mathbf{f}$) |
| 2:    $\widehat{\mathbf{x}} \leftarrow$ SCAN($\mathbf{x}$)     ▷ Speculative scan. | 2:    $(\widehat{\mathbf{x}}, \widehat{\mathbf{f}}) \leftarrow$ (SCAN($\mathbf{x}$), SCAN($\mathbf{f}$)) |
| 3:    $\mathbf{f}^- \leftarrow \mathbf{f}(1:)$ appended with 1 | 3:    $\mathbf{f}^- \leftarrow \mathbf{f}(1:)$ appended with 1 |
| 4:    $\mathbf{z} \leftarrow$ COMPRESS $(\widehat{\mathbf{x}}, \mathbf{f}^-)$ | 4:    $\mathbf{w} \leftarrow$ COMPRESS($\widehat{\mathbf{x}}, \mathbf{f}^-$) |
| 5:    **return** DIFF($\mathbf{z}$) | 5:    **return** REVERT($\widehat{\mathbf{x}}, \widehat{\mathbf{f}}, \mathbf{w}$) |

To the best of our knowledge, the derivation of segmented sum as a composition of SCAN, COMPRESS, and vector DIFF, has not been explicitly discussed in the literature, but similar ideas exist. One relevant example is the "fast segmented sum", Algorithm 6 of (Liu & Vinter, 2015b), which we call FSS from now on. Similarly to SCD, FSS performs an unsegmented scan on $\mathbf{x}$. FSS requires an additional copy of the input data vector to a temporary vector. Next, using the additional temporary vector, FSS computes the segmented sums using a gather in a parallel for-loop. In contrast, and crucially, SCD does not require an additional temporary vector and could take advantage of the MMU for the index calculations of the parallel compaction/compress operation.

The Segmented Scan Algorithm 4.2, dubbed SSCR (Scan, Scan, Compress, and Revert), shares a similar approach to the SCD algorithm. First, it performs a speculative scan of $\mathbf{x}$ and then a parallel compress to collect the last entries of each segment. In addition, SSCR requires a scan of the boolean flag vector along with a parallel correction operation that could be performed in the VCU (Line 5 of Algorithm 4.2). We call the vector-only correction operator *REVERT*, which takes as input $\widehat{\mathbf{x}}, \widehat{\mathbf{f}}$, and $\mathbf{w}$, and it subtracts from each segment the last element of the previous segment. Crucially, we can show that there exists an AC[0] circuit with size $\mathcal{O}(n^2)$ which performs the REVERT operation, leading to the following theorem. The full proof is detailed in Appendix A.3.5.

**Theorem 4.2** (SCD & SSCR). *In MMV-RAM, SCD and SSCR (Algorithms 4.1 and 4.2) can be implemented such that they return the segmented sum and scan of $\mathbf{x}$ over $\mathbf{f}$, respectively, using:*

$$\mathcal{O}(\log_s(n)) \text{ steps}, \ \mathcal{O}\left(\mathcal{M}(\tfrac{n}{s}) + nB(n+B)\right) \text{ work},$$

*and $B \in \mathcal{O}(\log(nM))$ bits-per-element to avoid overflow, where $M := \max_{i \in [n]\}} |x(i)|$. Any vector-only MMV-RAM algorithm requires $\Omega\left(\frac{\log(n)}{\log\log(n)}\right)$ steps.*

**Improving the work complexity.** Theorem 4.2 states that Algorithms 4.1 and 4.2 achieve a theoretical speed-up against any VCU-only MMV-RAM algorithm, for appropriate values of $s$. However, their work scales quadratically with respect to $n$, which is not ideal. In Appendix A.3.6 we show that we can improve the work to $\mathcal{O}(n)$, while maintaining the same step complexity. To achieve this the analysis is rather involved. We need to follow a different approach, which requires specialized circuitry that might not be available on existing hardware. As such, at least at the time of this writing, this improvement is mainly of theoretical interest.

In brief, the methodology follows a block-recursive approach to compute segmented scans. The core idea of computing segmented scans (recursively) via a contraction argument is well-known in the literature; see (Blelloch, 1989) and (Sengupta et al., 2007, Algorithms 3 and 4) for circuits with fan-in two. (Zouzias & McColl, 2023) extended the contraction approach to blocks (or fan-in) of size $s$ for unsegmented scan. Our approach is a generalization for the segmented case. A careful analysis yields the following Theorem 4.3, whose proof is detailed in Appendix A.3.7. Intriguingly, our analysis shows that, even though segmented scan is not in AC[0], if we have black-box access to a full (unsegmented) scan, the speculation can be corrected with an AC[0] circuit!

**Theorem 4.3.** *Given a value vector* $\mathbf{x}$*, where* $|\mathbf{x}(i)| \leq M$ *for some* $M \in \mathbb{Z}_{\geq 0}$*, and a boolean vector* $\mathbf{f}$*, both of size* $n$*, there exists an MMV-RAM algorithm which computes the segmented scan* $\mathbf{z}$ *of* $\mathbf{x}$ *over* $\mathbf{f}$ *using* $\mathcal{O}(\log_s(n))$ *steps and* $\mathcal{O}\left(\mathcal{M}(\frac{n}{s}) + nB(s + \frac{B}{s})\right)$ *work, where* $B \in \mathcal{O}\left(\log(nM)\right)$ *bits-per-element are sufficient to avoid overflow.*

### 4.2 Applications of segmented operations

We next discuss higher-level algorithms in the MMV-RAM model, by using segmented operations as building blocks. The problems studied are integer / element-wise vector / matrix multiplication, and SpMV. Our results are summarized in Table 4.1. Their analysis is provided in Appendix A.4.

The first application is the product of two integers, and the element-wise product between two vectors with integer elements. We remind that the product of two $B$-bit integers is *not* in AC[0], due to parity (Vollmer, 1999). From the lower bound of Theorem 2, this means that any boolean circuit for integer multiplication $B^{\mathcal{O}(1)}$ size will have $\Omega\left(\frac{\log(B)}{\log\log(B)}\right)$ depth. Using segmented scan as building block, we can exploit the MMU to achieve a circuit of smaller depth and polynomial size, as described in Lemma A.3 in the appendix. Generalizing this lemma, we can efficiently compute the element-wise multiplication between two vectors (Corollary A.1). The second application builds

Table 4.1: Applications of segmented operations. $M \in \mathbb{Z}_{\geq 0}$ upper bounds the input magnitudes.

| Problem | Steps | Work | Word-length ($B$) | Steps without MMU | Proof |
|---|---|---|---|---|---|
| Integer product | $\mathcal{O}(\log_s(B))$ | $\mathcal{O}\left(\mathcal{M}(\frac{B}{s}) + sB^2\right)$ | $\mathcal{O}(\log(M))$ | $\Omega\left(\frac{\log(B)}{\log\log(B)}\right)$ | Lem. A.3 |
| Element-wise vector product | $\mathcal{O}(\log_s(B))$ | $\mathcal{O}\left(\mathcal{M}(\frac{nB}{s}) + nsB^2)\right)$ | $\mathcal{O}(\log(nM))$ | $\Omega\left(\frac{\log(B)}{\log\log(B)}\right)$ | Cor. A.1 |
| Matrix product | $\mathcal{O}(\log_s(nB))$ | $\mathcal{O}\left(\mathcal{M}\left(\frac{n^3B}{s}\right) + n^3sB^2\right)$ | $\mathcal{O}(\log(nM))$ | $\Omega\left(\frac{\log(n)}{\log(\log(n))}\right)$ | Thm. A.2 |
| SpMV | $\mathcal{O}(\log_s(nB))$ | $\mathcal{O}\left(\mathcal{M}(\frac{nB}{s}) + n^2B + nsB^2\right)$ | $\mathcal{O}(\log(nM))$ | $\Omega\left(\frac{\log(n)}{\log(\log(n))}\right)$ | Thm. A.3 |

on top of element-wise vector multiplication to design an MMV-RAM algorithm for the product of two integer matrices $\mathbf{A}$ and $\mathbf{B}$. In Theorem A.2 it is proved that it is possible to execute the matrix product $\mathbf{AB}$ in MMV-RAM using $\mathcal{O}(\log_s(n))$ steps and $\mathcal{O}(n^3B)$ work, where $B$ is the number of bits-per-element. However, if we use only the VCU to perform the sums (reductions) in the final step of a bilinear matrix multiplication algorithm, we need to execute $\Omega(\log(n)/\log\log(n))$ steps, due to parity. This still holds even if we get the initial scalar multiplications "for free".

Finally, we discuss the application of our SCD algorithm for SpMV. The resulting Algorithm 4.3, is based on the well-known segmented sum SpMV approach (Blelloch et al., 1993; Liu & Vinter, 2015b). It has the benefit that

---

**Algorithm 4.3** CSR SpMV in MMV-RAM.

1: **procedure** SPMV($\mathbf{A}$, $\mathbf{x}$)                    ▷ $\mathbf{A}$ is in CSR format
2:     $\mathbf{w} \leftarrow$ GATHER($\mathbf{x}$, $\mathbf{A}$.col)         ▷ $\mathbf{w}(i) = \mathbf{x}(\mathbf{A}.\text{col}(i))$
3:     $\mathbf{z} \leftarrow$ MULT($\mathbf{w}$, $\mathbf{A}$.val)         ▷ Element-wise product
4:     **return** SCD($\mathbf{z}$, $\mathbf{A}$.row)

---

two out of its three subroutines, MULT and SCD, make extensive use of the MMU for speculative scan and compression operations. The complete analysis of the algorithm is described in Appendix A.4.3. We note that in Step 4, we used an integer vector ($\mathbf{A}.\text{row}$) which corresponds to the segment start indices, instead of the boolean flags that SCD expects by default. The complexity is unaffected.

## 5 EXPERIMENTS

In this section we provide an experimental evaluation of our proof-of-concept implementations of the proposed MMV-RAM algorithms, using the *Ascend AI 910B* accelerator as a case study. The architecture of Ascend accelerators, namely the DaVinci architecture, combines MMUs, called Cube cores, with VCUs that offer SIMD capabilities, and scalar processors (Liao et al., 2019; 2021) (see Figure A.2 in the Appendix). Additionally, each core is linked with Memory Transfer Engines (MTE) that transfer data between global / local memory buffers, and a hierarchy of intermediate scratchpad memories to store intermediate tiled data.

**Evaluation setup.** All proposed algorithms were implemented in C++17 using the AscendC programming framework (cf. Appendix A.5.2). We used the Ascend CANN toolkit 8.0.RC2.alpha003 with Ascend firmware and drivers versions 1.0 and 23.0.0, respectively. All evaluations are performed on the Ascend 910B4 accelerator (Liao et al., 2019; Huawei Cloud AI Team, 2025) containing 20 Cube and 40 Vector cores, and an AMD EPYC CPU processor, running on Ubuntu 22.04. Our custom AscendC segmented operators are integrated in PyTorch with *pybind11* and PyTorch C++ extension functionalities, namely, using the PyTorch Ascend adapter v2.4.0 (open-sourced at https://gitee.com/ascend/pytorch) to report our PyTorch-related measurements.

**Remark 5.1.** *Although we focus on Ascend, a similar experimental evaluation could be performed with any architecture that contains both MMUs and VCUs, for examples, the ones listed in Table 3.1.*

**Segmented scan using a single AI-core.** Here, we measure the performance benefits of using the Cube core (MMU) of the Ascend accelerator for segmented scans. We implemented a single-AI-core algorithm based on the speculative block-scan ideas of Algorithm 4.2 (SSCR) and the more advanced Algorithm A.2 from Theorem 4.3. As a first step, the implementation uses the Cube core to perform length-$s$ consecutive scans. We set $s = 128$ to maximize the L0A/L0B scratchpad memories utilization. In the second step, these speculative block scans are transferred to the VCU to correct the miss-speculations (cf. Appendix A.3.6).

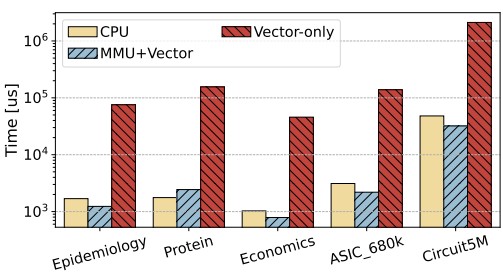

Figure 5.1: Segmented scan (single AI-core) vs. Vector-only and single-thread CPU (optimized with -O3) implementations.

To highlight the benefits of utilizing the Cube core, we also implement a *vector-only* algorithm that performs the speculative scan within the VCU, using the cumsum AscendC API, and then corrects the speculation. We call the baseline method "Vector-only". For reference, we also report the performance of a C++ CPU implementation using a vanilla for-loop segmented scan.

Figure 5.1 illustrates the performance comparison of the aforementioned implementations. We chose five different matrices from various scenarios, from epidemiology to economics. Such matrices have already been chosen from previous related works (Liu & Vinter, 2015b), and their dimensions span from $36K \times 36K$ up to $5M \times 5M$ (details in Table A.3 in Appendix A.5.3). The MM+V version is significantly faster than the Vector-only baseline. Notably, the MM+V approach reaches (and even slightly surpasses) the CPU performance, which is included for normalization purposes. These observations align with the analysis of Section 3, where we argued that, in MMV-RAM, a speed-up is expected when using both MMUs and VCUs, particularly for larger values $s$.

**Sparse Matrix-Vector Multiplication.** We finally evaluate our algorithms for SpMV, which is arguably the most interesting application. Figure 5.2 illustrates the performance of our SpMV implementation, based on Algorithm 4.3. As there is currently no optimized SpMV implementation for Ascend architectures, we compare to well-established CPU implementations as baselines, from the Intel Math Kernel Library (MKL) and Eigen (Guennebaud et al., 2010).

The goal of this comparison is not to claim superiority in performance, but rather to report how a proof-of-concept implementation of our algorithms compares against highly-optimized implementations (both vendor and open-source). To that end, the matrix sizes are deliberately chosen up to $\sim 70K \times 70K$, so that the right-hand-side vectors fit in the L3 cache of the CPU, and in the Ascend

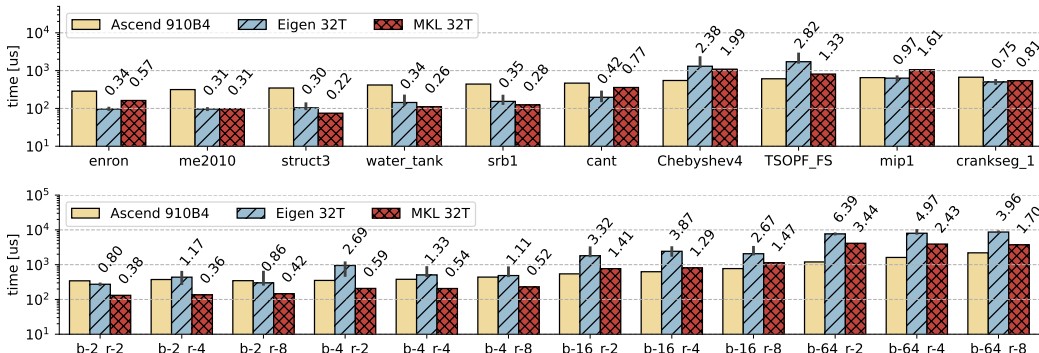

Figure 5.2: SpMV runtime comparison (median of 10 runs with 95% confidence intervals) for matrices from the SuiteSparse collection (Davis & Hu, 2011) (top), and synthetic sparse-attention matrices (Zaheer et al., 2020) (bottom), in ascending order based on the number of nonzeros.

scratchpad memory, respectively. This allows us to better study the effect of exploiting MMUs in the kernels with respect to the arithmetic intensity, rather than data movements between the higher cache-levels (I/O complexity and external memory algorithms are not considered here). Each bar is the median of ten consecutive runs. We report 32-thread CPU runtimes for reference.

At the top of Figure 5.2, we consider several representative datasets from (Davis & Hu, 2011), with varying non-zero densities and sparsity structures, and from different application domains, including networks, optimization, fluid dynamics, and structural problems (more details can be found in Table A.4 in the Appendix). At the bottom of Figure 5.2 we use synthetic matrices which follow the (blocked) sparse-attention sparsity structure, which was pioneered in the seminal work of (Zaheer et al., 2020). All these matrices have $65,536$ rows and columns, two blocks of "global attention", and three blocks of "window attention". We report the performance for all combinations of block sizes in $\{2, 4, 16, 64\}$ and number of random attention in $\{2, 4, 8\}$ (the matrix naming convention b-X_r-Y means block size equal to X and Y random attention blocks).

For all datasets, the MMU-based implementation performs comparably with the (optimized) CPU counterparts, and it even achieves noteworthy speed-ups for several cases, up to $3.44\times$ (MKL) and $6.39\times$ (Eigen) for the sparse-attention matrix b-64_r-2, which corresponds to the standard block size / random blocks that are typically used in existing models (including the original work of Zaheer et al. (2020)). Notably, the MMU-based implementation demonstrates near-linear scaling with the number of non-zeros, showcasing robustness against irregular sparsity structures. These observations are encouraging towards exploiting (dense) MMUs for sparse computations in LLMs and other domains, but their full capabilities are yet to be explored. For additional experiments we refer to Appendix A.5.4, where we also provide a performance breakdown analysis of the subroutines used by our main algorithms.

## 6 CONCLUSION

In this work we provided novel insights and theoretical foundations for the design, analysis, and implementation of algorithms on AI accelerators. We proposed MMV-RAM, a machine model for such accelerators, which extends the Vector-RAM model with matrix multiplication processing units. In MMV-RAM, we designed and analyzed algorithms for fundamental operations (segmented scan and sum, matrix product, and SpMV), that use both VCUs and MMUs. By exploiting lower bounds from circuit complexity, we showed provable theoretical speed-ups against VCU-only MMV-RAM algorithms, and provided explicit trade-offs between the size of the MMU (parameter $s$ of MMV-RAM) and the achievable speed-ups. The key algorithmic idea is to perform *speculative* blocked computations (scans) using the MMU, which are then carefully corrected using the VCU. The proposed algorithms were implemented and experimentally evaluated using the Ascend 910B AI accelerator. Our proof-of-concept implementation demonstrates significant speed-ups against a vector-only baselines for segmented operations, and comparable performance with highly-optimized SpMV libraries, even outperforming them for certain sparse-attention datasets.

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

# A APPENDIX

## A.1 EXTENSIONS OF MMV-RAM

Here we briefly discuss possible variants and extensions of the MMV-RAM model.

### A.1.1 DIFFERENT CIRCUIT CLASSES FOR THE VCU

The VCU of MMV-RAM consists of AC[0] circuits. Other circuit complexity classes might be worth investigating, and what is their effect in the analysis of algorithms. As an example, recall that our main algorithms for segmented operations rely on gates with fan-in and fan-out of $\mathcal{O}(\text{poly}(s, B))$ size, i.e., independent of input size. Some hardware implementations, however, might only support constant fan-in gates. As such, NC[0] can be investigated as an alternative for the VCU. Other classes can also be considered depending on the applications. Some other examples from (Vollmer, 1999) include semi-bounded fan-in circuits (SAC), AC "with counters" (ACC), or AC[0] circuits extended with additional modulo gates. These classes lie between NC[0] and NC[1] in the so-called NC-hierarchy (cf. (Vollmer, 1999, Chapter 4)). In all cases, the complexity of VCU instructions (size-depth) and the balance with the MMU should be carefully calibrated.

### A.1.2 EXTENDING THE RIGHT-HAND-SIDE OF THE MMU

Given two input matrices $\mathbf{A}$ and $\mathbf{B}$ with sizes $n \times s$ and $s \times s$, the MMU of MMV-RAM computes $\mathbf{C} = \mathbf{A}\mathbf{B}$ in a single step. There is a clear asymmetry between $\mathbf{A}$ and $\mathbf{B}$. If we partition $\mathbf{A}$ into blocks of size $s \times s$, i.e., $\mathbf{A} = \begin{pmatrix} \mathbf{A}_1^\top & \mathbf{A}_2^\top & \mathbf{A}_2^\top & \ldots \end{pmatrix}^\top$, then each block $\mathbf{A}_i$ is multiplied with the same matrix $\mathbf{B}$ from the right. It is reasonable to ask whether we can build a more powerful (and/or more "symmetric") model by allowing each block $\mathbf{A}_i$ to be multiplied with a different matrix $\mathbf{B}_i$ from the right. For example, assuming a total of $k$ blocks of size $s \times s$, the MMU now computes

$$\begin{pmatrix} \mathbf{A}_1\mathbf{B}_1 \\ \vdots \\ \mathbf{A}_k\mathbf{B}_k \end{pmatrix} \leftarrow \begin{pmatrix} \mathbf{A}_1 \\ \vdots \\ \mathbf{A}_k \end{pmatrix} \odot \begin{pmatrix} \mathbf{B}_1 \\ \vdots \\ \mathbf{B}_k \end{pmatrix},$$

instead of

$$\begin{pmatrix} \mathbf{A}_1\mathbf{B} \\ \vdots \\ \mathbf{A}_k\mathbf{B} \end{pmatrix} \leftarrow \begin{pmatrix} \mathbf{A}_1 \\ \vdots \\ \mathbf{A}_k \end{pmatrix} \mathbf{B}.$$

In this modification, the MMU acts as a "block-vector unit", executing small matrix multiplications between blocks. Such an extension seems particularly interesting, for example, for large matrix multiplications. Indeed, assuming any matrix multiplication algorithm that can be written as a bilinear circuit, the scalar product gates can be replaced with block-wise multiplications of size $s \times s$. However, our efforts so far did not reveal any significant advantages of this extension over the proposed MMV-RAM definition of Section 3. The main reason is the following. The (depth) hardness of matrix multiplication arises from the parity problem: in any bilinear circuit for matrix multiplication, all scalar multiplications can be executed in constant depth, but the reductions of these scalar products to obtain the bilinear forms necessarily require a logarithmic number of steps (if the circuit size remains polynomial). Our Theorem A.2 indicates that we can perform (large) matrix multiplications in $\mathcal{O}(\log_s(n))$ steps in MMV-RAM. On the other hand, in the extended model, the number of steps can potentially be reduced to $\mathcal{O}(\log_s(n/s))$, which barely saves a constant number of steps. Based on these observations, the possible benefits of this seemingly natural model extension are yet to be clarified, and left as future research.

## A.2 MATRIX MULTIPLICATION CIRCUITS

In this appendix, we provide the corresponding background and existing results for matrix multiplication circuits, which are outlined in Table A.1.

We first recall two arithmetic circuit sub-classes from (Raz & Shpilka, 2001; Raz, 2002). *Bounded-coefficient* arithmetic circuits impose the restriction that all weights and inputs of product gates have

Table A.1: Existing upper and lower bounds for the size of constant-depth, unbounded fan-in circuits for the product of two $s \times s$ matrices.

| Gates | Depth | Size upper bound | Size lower bound | Input | Notes |
|---|---|---|---|---|---|
| $\{+, \times\}$ | $d = 2$ | $\mathcal{O}(s^3)$ | $\Omega(s^3)$ (Raz & Shpilka, 2001) | Real numbers | Textbook algorithm |
| $\{\wedge, \vee, \neg\}$ | $d = \mathcal{O}(1)$ | $s \exp(s)$ | $\exp(\Omega(s^{\frac{1}{d-1}}))$ (Håstad, 1986) | $\mathcal{O}(\log(s))$-bit integers | See also Section A.2 |
| $\{\wedge, \vee, \neg, \text{Th}\}$ | $\mathcal{O}(d), d \geq 1$ | $\widetilde{\mathcal{O}}(ds^{a+\beta\gamma^d})$ (Parekh et al., 2018) | $\Omega\left(s^2 \frac{\lambda_d(s^2)}{d^2}\right)$ (Raz & Shpilka, 2001) | $\mathcal{O}(\log(s))$-bit integers | $a = \log_2(7), \beta \approx 1.6,$ $\gamma \approx 0.5$ |

magnitude at most one. *Bilinear* arithmetic circuits consist of three subcircuits $\Sigma_1, \Pi$, and $\Sigma_2$. The subcircuit $\Sigma_1$ receives as inputs the elements of the two matrices and contains only $\{+\}$ gates (of unbounded fan-in). The second subcircuit, $\Pi$, consists of only one level of $\{\times\}$ gates with fan-in two. The inputs of these $\{\times\}$ gates are the outputs of $\Sigma_1$. Finally, $\Sigma_2$ contains only $\{+\}$ gates, which receive as inputs the outputs of $\Pi$. The outputs of $\Sigma_2$ are the elements of the matrix product.

**Arithmetic circuits.** The arithmetic complexity of multiplying two $s \times s$ matrices (in terms of scalar additions and multiplications) is $\mathcal{O}(s^{\omega+\eta})$, for any $\eta > 0$, where the currently best known upper bound for the exponent is $\omega < 2.371339$ (Alman et al., 2025). Depth-size trade-offs for general arithmetic circuits were thoroughly studied by Raz and Shpilka (Raz & Shpilka, 2001). The authors proved that any matrix multiplication arithmetic circuit with size $S$ and depth $d$ can be transformed into an equivalent bilinear circuit of size $\mathcal{O}(S)$ and depth $\mathcal{O}(d)$, and that any bilinear circuit whose longest path from the product gates to the outputs has length $d \leq 1$ necessarily has size $\Omega(s^3)$. The standard dot-product based matrix multiplication algorithm achieves this size and depth. For arbitrary $d \geq 2$, the authors proved a $\Omega(\frac{\lambda_d(s^2)}{d^2}s^2)$ size lower bound, where $\lambda_d(s^2)$ is a family of slow-growing functions, in particular, $\lambda_2(x) = \log(x)$, $\lambda_3(x) = \log(\log(x))$ (see (Raz & Shpilka, 2001) for more details). Raz (Raz, 2002) proved an $\Omega(s^2 \log(s))$ lower bound for the size of any bounded-coefficient arithmetic circuit for matrix multiplication, including depth-size tradeoffs.

**Boolean circuits.** For the multiplication of integer matrices using (unbounded fan-in) boolean circuits, depth lower bounds can be directly obtained from the parity problem (Furst et al., 1984; Yao, 1985; Håstad, 1986). To see this, note that a sequence $X \in \{0, 1\}^n$ can be copied to the first row of a $\{0, 1\}^{n \times n}$ matrix $\mathbf{A}$. Then one can compute the product $\mathbf{AB}$ where $\mathbf{B}$ is the all-ones matrix. The least significant bit of the top-left element of the result gives the parity of $X$. This means that any constant-depth matrix multiplication boolean circuit necessarily has exponential size, which can be achieved, for example, by "brute-forcing" the corresponding DNF.

**Threshold circuits.** Threshold circuits were thoroughly analyzed in (Parekh et al., 2018). Focusing on Strassen's algorithm ((Strassen, 1969)) as a baseline, the authors proved several size-depth trade-offs. One of the main contributions that we import here is a TC[0] circuit with size $\mathcal{O}(s^{3-\delta})$ for some $\delta \in (0, 1)$ to multiply two $s \times s$ matrices with $\mathcal{O}(\log(s))$-bit integer elements. It is also outlined how to extend the analysis for other fast matrix multiplication algorithms besides Strassen's. Size lower bounds can be obtained from (Raz & Shpilka, 2001) for this case as well. The aforementioned $\Omega(\frac{\lambda_d(s^2)}{d^2}s^2)$ size lower bound of (Raz & Shpilka, 2001) also holds for boolean circuits with arbitrary gates (including threshold circuits) for matrix multiplication over GF[2]. The latter directly reduces to integer matrix multiplication, which gives the lower bound.

Finally, to simulate rectangular matrix multiplications of size $m \times s$ and $s \times s$, for arbitrary $m$, there are many different options. The simplest is to assume $m/s$ copies of a square matrix multiplication circuit of size $s \times s$, such as the standard inner-product algorithm or the TC[0] circuits of (Parekh et al., 2018). Alternatively, one can consider circuits dedicated to rectangular matrix multiplications (recall that two matrices of size $s \times s^\alpha$ and $s^\alpha \times s$ can be multiplied in $\mathcal{O}(s^2)$ arithmetic operations for all $\alpha \lesssim 0.32$ (Alman et al., 2025; Williams et al., 2024)). Other specialized circuits can be used, for example, if the matrices have very large entries (Harvey & Van Der Hoeven, 2018).

To conclude this section, we can mention the following works regarding other aspects of matrix multiplication algorithms. (McColl & Tiskin, 1999) provides a detailed analysis in the BSP model.

The numerical stability of fast matrix multiplication algorithms in the floating point model was studied in (Demmel et al., 2007). Communication and I/O complexity have been analyzed in (Hong & Kung, 1981; Ballard et al., 2012; Solomonik & Demmel, 2011). Finally, we refer to the recent work of (Alman & Yu, 2025), which heavily reduces the leading constants of fast matrix multiplication algorithms.

### A.3 Deferred proofs and algorithms

In this appendix we provide additional proofs and analysis of algorithms which were deferred from the main text.

#### A.3.1 Proof of Theorem 3.1 (balance between MMU and VCU)

First, regarding RELU, note that the MMU of MMV-RAM can only execute linear transformations. RELU is not a linear function, and therefore it cannot be implemented using the MMU. It might be able to approximate it, but this is not covered in this work, since RELU can easily be implemented with the VCU, which is also the case in existing architectures. For the other two items in Theorem 3.1, we first recall the following classic theorem by Håstad.

**Theorem A.1** (Restatement of Håstad (1986), Thm. 1). *For every $s > s_0^d$, where $s_0$ is a constant, the size $L$ of a depth-$d$ boolean circuit that computes the parity of $s$ bits has size $L > 2^{(0.1)^{\frac{d}{d-1}} s^{\frac{1}{d-1}}}$.*

Solving for $d$ gives the following trade-off:

$$d > \frac{\log(s)}{\log(\log(L)) + c_1} + c_2, \qquad (2)$$

where $c_1$ and $c_2$ are constants. This allows us to prove a clear separation between the computational power of the MMU and the VCU in the MMV-RAM model. A straightforward example is the following.

**Fact A.1.** *Let $\mathbf{A}$ and $\mathbf{B}$ be two $B$-bit integer matrices with sizes $n \times s$ and $s \times s$, respectively. Let $\mathcal{A}$ be an algorithm in the MMV-RAM model that uses only the scalar and the vector units to compute the matrix product $\mathbf{AB}$. If the work of $\mathcal{A}$ is upper bounded by $s^{O(1)}$, then $\mathcal{A}$ executes $\Omega\left(\frac{\log(s)}{\log(\log(s))}\right)$ steps.*

*Proof.* If we want to compute the parity of an $s$-bit sequence, we can place the $s$ bits as the first row of $\mathbf{A}$, and set $\mathbf{B}$ to be the all-ones matrix. The least significant bit of the top-left element of the product $\mathbf{AB}$ gives the parity. Since $\mathcal{A}$ only uses the scalar and vector units, it forms an unbounded fan-in Boolean circuit, and therefore Eq. (2) gives the result. $\square$

We finally show that the MMU is not strong enough by itself even for linear transformations. In particular, we argue that even if the VCU implements a "weaker" class of circuits, it still provides crucial capabilities to the model. One example is the following.

**Lemma A.1.** *Let $s$ be the MMV-RAM model parameter, and let $m$, $n$ be two positive integers such that $n = ms$, and $m \gg s$. There exists a vector operation which can be efficiently implemented in the VCU, i.e., with an AC[0] circuit of size $n$ and depth $\mathcal{O}(1)$, but it requires $\Omega(s)$ calls to the MMU.*

*Proof.* In the context of this lemma, only two types of operations are allowed: reading and writing consecutive positions between the memory and the MMU, and executing left and right matrix multiplications. In order to model them algebraically, we first introduce some notation.

We always treat the vector $\mathbf{x}$, which has length $n$ initially, as the contents of the memory. Using a single read operation, we can load any $ks$ consecutive elements of $\mathbf{x}$ in the left "matrix register" of the MMU, for some $k$, starting from some arbitrary position $r$, or $s^2$ consecutive positions in the right register of the MMU. In the former case, the $ks$ consecutive elements of $\mathbf{x}$ are viewed as a row-major matrix of size $k \times s$. We now want to multiply $\mathbf{X}$ with some arbitrary $s \times s$ matrix $\mathbf{B}_s$

from the right. By using the mixed-product property of the Kronecker product $\otimes$, this operation can be written as follows:

$$\text{Right multiplication: } \mathbf{x}' \leftarrow \begin{pmatrix} \mathbf{I}_{r-1} & & \\ & \mathbf{I}_k \otimes \mathbf{B}_s^\top & \\ & & \mathbf{I}_{n-ks-r+1} \end{pmatrix} \mathbf{x}.$$

In a similar way we model the left multiplications. In this case we read $s^2$ consecutive elements of $\mathbf{x}$, and store them as a row-major $s \times s$ matrix $\mathbf{X}$ in the right register of the MMU, starting again at some arbitrary position $r$. $\mathbf{X}$ needs to be multiplied from the left with some other matrix $\mathbf{A}_{k \times s}$ with size $k \times s$, and then the resulting matrix $\mathbf{A}_{k \times s}\mathbf{X}$ is stored back in memory. Using again properties of Kronecker products, we can write:

$$\text{Left multiplication: } \mathbf{x}' \leftarrow \begin{pmatrix} \mathbf{I}_{r-1} & & \\ & \mathbf{A}_{k \times s} \otimes \mathbf{I}_s & \\ & & \mathbf{I}_{n-s^2-r+1} \end{pmatrix} \mathbf{x},$$

Having expressed the basic operations as matrix-vector products, we can now derive lower bounds. Let us study how to perform the following block-reordering operation of a large vector. Assume a vector $\mathbf{x}$ of size $n = ms^2$, for some $m$, which is partitioned in $ms$ blocks of size $s$ each, and the blocks are grouped in consecutive groups of $s$ blocks each. The goal is to reverse the order of the blocks within each group. Algebraically, this operation can be written as follows:

$$\mathbf{y} \leftarrow (\mathbf{I}_m \otimes \mathbf{J}_s \otimes \mathbf{I}_s)\,\mathbf{x},$$

where $\mathbf{J}_s$ is the identity matrix with its columns in reverse order. Note that this reordering can easily be implemented with an AC[0] circuit.

Assume that we have an MMV-RAM algorithm which performs this reordering using only the aforementioned operations. Based on the above, this algorithm can be written as a sequence of matrix-vector products

$$\mathbf{y} \leftarrow \mathbf{L}_1 \mathbf{R}_1 \mathbf{L}_2 \mathbf{R}_2 \ldots \mathbf{L}_k \mathbf{R}_k \mathbf{x},$$

where $\mathbf{L}_i$ and $\mathbf{R}_i$ are left and right multiplication operations as defined above (some of them might be equal to the identity, in which case these operations are ignored). Now we will derive an $\Omega(s)$ lower bound on the number of operations that need to be executed.

First, note that the equality holds necessarily:

$$\mathbf{L}_1 \mathbf{R}_1 \mathbf{L}_2 \mathbf{R}_2 \ldots \mathbf{L}_k \mathbf{R}_k = \mathbf{I}_m \otimes \mathbf{J}_s \otimes \mathbf{I}_s,$$

because the transformation must return the correct result for every input vector. Therefore, it suffices to argue about the minimum number of $\mathbf{L}$ and $\mathbf{R}$ factors required to produce the matrix $\mathbf{I}_m \otimes \mathbf{J}_s \otimes \mathbf{I}_s$.

We can now observe that a sequence of transformations $\mathbf{R}_1 \mathbf{R}_2 \ldots \mathbf{R}_k$ cannot produce the matrix $\mathbf{I}_m \otimes \mathbf{J}_s \otimes \mathbf{I}_s$ with less than $k = \Omega(s)$ products. This is because the matrices $\mathbf{R}_i$ are banded with bandwidth $s-1$, therefore, each product increases the bandwidth by at most $s$. However, the matrix $\mathbf{I}_m \otimes \mathbf{J}_s \otimes \mathbf{I}_s$ has a bandwidth of $s^2$.

Next, note that, due to its structure, a transformation $\mathbf{L}_i$ operates on at most $s^2$ consecutive elements of $\mathbf{x}$. Assume now that we have executed a sequence of transformations $\mathbf{L}_1 \mathbf{R}_1 \mathbf{L}_2 \mathbf{R}_2 \ldots \mathbf{L}_k \mathbf{R}_k \mathbf{x}$. Based on the previous argument, there are at least $n - ks^2$ elements of $\mathbf{x}$ that have only been modified by the $\mathbf{R}_i$ transformations. If $n$ is large enough, this means that there is at least an entire block $\mathbf{J}_s \otimes \mathbf{I}_s$ which has has not been modified by $\mathbf{L}_i$ transformations. Based on the argument above, we need at least $k = \Omega(s)$ left and right transformations to reorder this block, which concludes the proof.

$\square$

### A.3.2 VCU INSTRUCTIONS

Table A.2 summarizes the vector instructions used for Algorithms A.1 and A.2. We recall that the latter achieves the best work for segmented scan (reported in Theorem 4.3). All these instructions can be implemented with uniform AC[0] circuits for every input size $n$ as follows.

- The standard operations AND/OR/NOT, can be implemented with size $n$, depth one, and constant fan-in/-out.

- MASK takes as input a vector $\mathbf{x}$ with integer elements, and a "mask" boolean vector $\mathbf{y}$, and copies the input element $\mathbf{x}(i)$ to the output position $\mathbf{z}(i)$ if $\mathbf{y}(i) = 1$, otherwise $\mathbf{z}(i)$ is set to zero. It can be achieved by performing a bit-wise AND of each $B$-bit element $\mathbf{x}(i)$ with $\mathbf{y}(i)$, in depth one, size $nB$, and constant fan-in/-out.

- ISZERO takes as input a vector with integer elements, and returns a boolean vector $\mathbf{z}$, such that $\mathbf{z}(i) = 1$ if the corresponding element $\mathbf{x}(i)$ is equal to zero, otherwise $\mathbf{z}(i) = 1$. This can be achieved with a vector of size $\mathcal{O}(nB)$, where we simply need to check that all the bits of each element $\mathbf{x}(i)$ are equal to zero, i.e., by negating each bit and using a $\vee$ gate with fan-in $\mathcal{O}(B)$ and fan-out one.

- Element-wise ADD/SUB can also be achieved with AC[0] circuits, with $\mathcal{O}(nB^2)$ and $\mathcal{O}(B)$ fan-in boolean gates. See e.g. (Vollmer, 1999, Chapter 1) or (Dhall, 1994, Chapter 8) for further details and for more advanced circuits and techniques. We note that, in the main algorithm, we only use subtractions $a - b$ for the special case where $a \geq b$.

- FILLS is parametrized by $s$. It copies the first element of each $s$-segment of the input vector $\mathbf{x}$, namely, $\mathbf{x}(is)$, for $i = 0, \ldots, \lceil \frac{n}{s} \rceil$, to the corresponding entire segment in the output vector, i.e., $\mathbf{z}(is : (i+1)s)$. The circuit has size $\mathcal{O}(nB)$, constant fan-in, and $\mathcal{O}(s)$ fan-out.

- SCATTERS/GATHERS are also parametrized by $s$. As the name suggests, GATHERS copies the last element of the $i$-th $s$-segment of $\mathbf{x}$, to $\mathbf{z}(i-1)$. Accordingly, SCATTERS copies the $(i-1)$-th element of $\mathbf{x}$ to the last position of the $i$-th $s$-segment of $\mathbf{z}$. The corresponding circuits have size $\mathcal{O}(sB)$, since they only copy one element per segment to a single position in the output, and the fan-in/-out of the gates are all constant.

- Finally, the instruction REVSPEC is a uniform circuit that corrects the misspeculated (unsegmented) scans in Algorithm A.2. This circuit is described in detail in Section 4, specifically, in Lemma A.2.

Table A.2: Circuit sizes and fan-in for vector operations of length $n$ used in the main Algorithm A.2. Integers (int) have bit-width $B$. All circuits have constant depth, and $s$ is the fixed model parameter. All instructions (except REVSPEC) are either present or easy-to-implement in existing architectures.

| | Operation | Size | Fan-in / -out | Description |
|---|---|---|---|---|
| 1. | $\mathbf{z} : \text{vec[bit]} \leftarrow \text{AND/OR}(\mathbf{x} : \text{vec[bit]}, \mathbf{y} : \text{vec[bit]})$ | $\mathcal{O}(n)$ | $\mathcal{O}(1)$ | $\mathbf{z}(i) \leftarrow \mathbf{x}(i) \wedge \mathbf{y}(i)$ (resp. $\mathbf{x}(i) \vee \mathbf{y}(i)$). |
| 2. | $\mathbf{z} : \text{vec[bit]} \leftarrow \text{NOT}(\mathbf{x} : \text{vec[bit]})$ | $\mathcal{O}(n)$ | $\mathcal{O}(1)$ | $\mathbf{z}(i) \leftarrow \neg\mathbf{x}(i)$. |
| 3. | $\mathbf{z} : \text{vec[int]} \leftarrow \text{MASK}(\mathbf{x} : \text{vec[int]}, \mathbf{y} : \text{vec[bit]})$ | $\mathcal{O}(nB)$ | $\mathcal{O}(1) / \mathcal{O}(B)$ | $\mathbf{z}(i) \leftarrow \mathbf{x}(i)$ if $\mathbf{y}(i) = 1$, else $\mathbf{z}(i) \leftarrow 0$. |
| 4. | $\mathbf{z} : \text{vec[bit]} \leftarrow \text{ISZERO}(\mathbf{x} : \text{vec[int]})$ | $\mathcal{O}(nB)$ | $\mathcal{O}(B) / \mathcal{O}(1)$ | $\mathbf{z}(i) \leftarrow 1$ if $\mathbf{x}(i) = 0$, else $\mathbf{z}(i) \leftarrow 0$. |
| 5. | $\mathbf{z} : \text{vec[int]} \leftarrow \text{ADD/SUB}(\mathbf{x} : \text{vec[int]}, \mathbf{y} : \text{vec[int]})$ | $\mathcal{O}(nB^2)$ | $\mathcal{O}(B)$ | $\mathbf{z}(i) \leftarrow \mathbf{x}(i) + \mathbf{x}(j)$ (resp. $\mathbf{x}(i) - \mathbf{x}(j)$). |
| 6. | $\mathbf{z} : \text{vec[int]} \leftarrow \text{FILLS}(\mathbf{x} : \text{vec[int]} ; s)$ | $\mathcal{O}(nB)$ | $\mathcal{O}(1) / \mathcal{O}(s)$ | $\mathbf{z}(is : is + s) \leftarrow \mathbf{x}(is), i \in \{0, \ldots, \lfloor \frac{n}{s} \rfloor\}$. |
| 7. | $\mathbf{z} : \text{vec[int]} \leftarrow \text{GATHERS}(\mathbf{x} : \text{vec[int]} ; s)$ | $\mathcal{O}(sB)$ | $\mathcal{O}(1)$ | $\mathbf{z}(i-1) \leftarrow \mathbf{x}(si-1), i \in \{1, \ldots, \lceil \frac{n}{s} \rceil\}$. |
| 8. | $\mathbf{z} : \text{vec[int]} \leftarrow \text{SCATTERS}(\mathbf{x} : \text{vec[int]} ; s)$ | $\mathcal{O}(sB)$ | $\mathcal{O}(1)$ | $\mathbf{z}(si-1) \leftarrow \mathbf{x}(i-1), i \in \{1, \ldots, \lceil \frac{n}{s} \rceil\}$. |
| 9. | $\mathbf{z} : \text{vec[int]} \leftarrow \text{REVSPEC}(\overline{\mathbf{x}} : \text{vec[int]}, \overline{\mathbf{f}} : \text{vec[int]} ; s)$ | $\mathcal{O}\left(nB(s^2 + \frac{B}{s})\right)$ | $\mathcal{O}(s + B) / \mathcal{O}(sB)$ | Reverts misspeculation (Lemma A.2). |

### A.3.3 PROOF OF THEOREM 4.1 (SCAN)

The following Algorithm A.1 achieves the result of Theorem 4.1.

---

**Algorithm A.1** Scan in MMV-RAM (simplified version of (Zouzias & McColl, 2023, Alg. 2))

---

1: **procedure** SCAN($\mathbf{x}; s$)                    $\triangleright n \leftarrow \text{len}(\mathbf{x})$
2:     $\mathbf{z} \leftarrow \text{MATMUL}(\mathbf{x}, \mathbf{U}_s)$
3:     **if** $n \leq s$ **then**
4:         **return** $\mathbf{z}$
5:     $\mathbf{x}_s \leftarrow \text{GATHERS}(\mathbf{z}; s)$
6:     $\mathbf{z}_s \leftarrow \text{SCAN}(\mathbf{x}_s; s)$
7:     $\mathbf{z} \leftarrow \text{SCATTERS}(\mathbf{z}_s; s)$
8:     $\mathbf{z}(s-1 : n) \leftarrow \text{MATMUL}(\mathbf{z}(s-1 : n), \mathbf{B}_s)$     $\triangleright \mathbf{B}_s$ is the identity with all-ones in the first row
9:     **return** $\mathbf{z}$

---

*Proof.* At each recursion step, the array size is reduced by a factor of $s$. Therefore, the recursion depth is $\mathcal{O}(\log_s(n))$. Regarding the cost of matrix multiplications, at each iteration $t = 1, \ldots, \lfloor \log_s(n) \rfloor$, the algorithm performs two MATMUL operations between two matrices with sizes of size $\mathcal{O}(n/s^t) \times s$ and $s \times s$, respectively. The cost of these operations is $\mathcal{M}(n/2^t)$ each. There are also three vector instructions per iteration: one SCATTERS, GATHERS. They both have work (circuit size) $\mathcal{O}(sB)$. We therefore have the following recursive formula for the work at iteration $t$:

$$
\begin{aligned}
W(t) &= W(t+1) + 2\mathcal{M}(\tfrac{n}{s^t}) + \mathcal{O}(sB) \\
&= \sum_{t=1}^{\lfloor \log_s(n) \rfloor} \mathcal{M}(\tfrac{n}{s^t}) + \mathcal{O}(sB \log_s(n)) \\
&\in \mathcal{O}\left( \mathcal{M}(\tfrac{n}{s}) + sB \log_s(n) \right).
\end{aligned}
$$

The total number of bits used to store numbers during the execution of the algorithm is bounded as follows. In iteration $t = 1, \ldots, \log_s(n)$, we have two MATMUL operations. These are the only operations that can increase the magnitude of the elements. Let $M(1) = M$, and denote by $M(t)$ the magnitude of the largest element at step $t$. Since the right-hand-side matrices have only zeros and ones, then every matrix multiplication can increase the magnitude of the input elements at most by a factor of $s$. Thus:

$$
M(t) \leq s^2 M(t-1) \in \mathcal{O}(s^{2\log_s(n)} M) = \mathcal{O}(n^2 M),
$$

which means that $\mathcal{O}(\log(n) + \log(M))$ bits are sufficient to avoid overflow. $\qquad\square$

### A.3.4 VECTOR DIFFERENTIATION USING MATRIX MULTIPLICATIONS

Here, we demonstrate that it is possible to use matrix multiplication to compute vector differentiation in the last step of Algorithm 4.1. Let $\mathbf{A}$ be a row-wise matrix view with $s$ columns of an arbitrary vector $\mathbf{x}$ (padded accordingly). Let

$$
\mathbf{D}_s := \begin{pmatrix}
1 & -1 & 0 & \ldots & 0 \\
0 & 1 & -1 & \ldots & 0 \\
0 & 0 & 1 & \ldots & 0 \\
\vdots & 0 & 0 & \ddots & -1 \\
0 & 0 & \ldots & 0 & 1
\end{pmatrix}_{s \times s}. \tag{3}
$$

It is easy to verify that the matrix product $\mathbf{A}\mathbf{D}_s$ computes the difference $x_i - x_{i-1}$ on each $s$-block except the boundary entries between blocks. The remaining entries on the boundary can be computed using the VCU.

### A.3.5 PROOF OF THM. 4.2 (SCD & SSCR)

*Proof.* We start with SCD (Algorithm 4.1). We first execute a full SCAN on $\mathbf{x}$ to obtain $\widehat{\mathbf{x}}$. Then, COMPRESS$(\widehat{\mathbf{x}}, \mathbf{f}^-)$ is implemented in two steps. In the first step, we perform a SCAN on $\mathbf{f}^-$. Let $\widehat{\mathbf{f}^-}$ be the result of this operation. Now, let $\mathbf{g} \leftarrow \mathsf{MASK}(\widehat{\mathbf{f}^-}, \mathbf{f})$. The non-zero elements of $\mathbf{g}$ contain the positions where the corresponding elements of $\widehat{\mathbf{x}}$ need to be written to form the vector $\mathbf{z}$, which is the final result of COMPRESS. To be able to perform such a "GATHER" step in constant depth, we can design an AC[0] circuit with all-to-all connections, i.e., size $\mathcal{O}(n^2 B)$, such that the element $\widehat{\mathbf{x}}(i)$ is written at position $j \leq i \leq N$ if and only if $\mathbf{g}(i) == j$. Given the vector $\mathbf{z}$, the DIFF operation can be straightforwardly implemented with an AC[0] circuit of size $\mathcal{O}(nB^2)$. The required number of bits is given by Theorem 4.1 for the SCAN operation.

The analysis is almost the same for SSCR (Algorithm 4.2). The first two steps are identical, since we are performing SCAN and COMPRESS. The only step that differs is the last one, which reverts the speculation. This can be implemented, for example, with an all-to-all communication as in the GATHER step above, with $\mathcal{O}(n^2 B)$ size and constant depth. $\qquad\square$

### A.3.6 BLOCK-RECURSIVE SEGMENTED SCAN ALGORITHM

In this appendix we show how to reduce the work complexity of Algorithms 4.1 and 4.2 from $\mathcal{O}(n^2)$ to only $\mathcal{O}(n)$, while maintaining the same step complexity. Before diving into the details, we give an overview of this more advanced algorithmic approach. Given an input vector $\mathbf{x}$ and a boolean vector $\mathbf{f}$ both of size $n$, the block-recursive approach initially partitions $\mathbf{x}$ and $\mathbf{f}$ into consecutive blocks of size $s$. Then the following steps are executed:

1. On each $s$-block of $(\mathbf{x}, \mathbf{f})$, compute (in parallel) the local scans of size $s$ using the MMU.

2. Use the VCU (e.g., the dedicated circuit of Lemma A.2) to revert the miss-speculated values, and store the results in place on $\mathbf{x}$.

3. Collect the values $(\mathbf{x}(s-1), \mathbf{x}(2s-1), \dots)$ into a vector $\mathbf{x}_s$ of size $\lceil \frac{n}{s} \rceil$ (zero-padded).

4. Create a vector $\mathbf{f}_s$ of size $\lceil \frac{n}{s} \rceil$ by applying logical OR to the elements of each $s$-block of $\mathbf{f}$.

5. Recursively compute the segmented scan of $(\mathbf{x}_s, \mathbf{f}_s)$ of size $\lceil \frac{n}{s} \rceil$, and store the result in $\mathbf{z}_s$.

6. On each $s$-block, update $\mathbf{x}$ using $\mathbf{z}_s$ by applying the associative binary operation between the last element of the previous block (skip the first block) and each element of the first segment of the current block.

Algorithm A.2 follows the aforementioned approach. It *speculatively* computes the "local" unsegmented scans of each block of size $s$. Then, the speculated scan will be updated through a specialized REVSPEC vector instruction, which corrects the misspeculated values to complete the local segmented scans.

---

**Algorithm A.2** Block-recursive segmented scan in the MMV-RAM model

1: **procedure** SEGSCAN($\mathbf{x}, \mathbf{f}; s$)                                  ▷ $s \geq 2$ is the **fixed** MMV-RAM parameter
2:      $\mathbf{z} \leftarrow$ BLOCKSEGSCAN($\mathbf{x}, \mathbf{f}; s$)
3:      **return** RECURSE($\mathbf{z}, \mathbf{f}; s$)
4: **procedure** RECURSE($\mathbf{z}, \mathbf{f}; s$)
5:      **if** $\text{len}(\mathbf{z}) \leq s$ **then**
6:          **return** $\mathbf{z}$
7:      $\bar{\mathbf{f}} \leftarrow$ MATMUL($\mathbf{f}, \mathbf{U}_s$)
8:      $\bar{\mathbf{f}}_s \leftarrow$ GATHERS $(\bar{\mathbf{f}}; s)$
9:      $\mathbf{f}_s \leftarrow$ NOT (ISZERO $(\bar{\mathbf{f}}_s)$)                              ▷ View flags in $\{0, 1\}$
10:     $\mathbf{z}_s \leftarrow$ GATHERS($\mathbf{z}; s$)
11:     $\mathbf{z}_s \leftarrow$ BLOCKSEGSCAN($\mathbf{z}_s, \mathbf{f}_s; s$)
12:     $\mathbf{z}_s \leftarrow$ RECURSE($\mathbf{z}_s, \mathbf{f}_s; s$)
13:     $\mathbf{z} \leftarrow$ SCATTERS($\mathbf{z}_s; s$)
14:     UPDATEFIRSTSEGMENT($\mathbf{z}(s-1:), \mathbf{f}(s-1:); s$)
15:     **return** $\mathbf{z}$
16: **procedure** BLOCKSEGSCAN($\mathbf{x}, \mathbf{f}; s$)
17:     $\bar{\mathbf{x}} \leftarrow$ MATMUL($\mathbf{x}, \mathbf{U}_s$)                          ▷ Speculative $s$-block scan
18:     $\bar{\mathbf{f}} \leftarrow$ MATMUL($\mathbf{f}, \mathbf{U}_s$)
19:     **return** REVSPEC $(\bar{\mathbf{x}}, \bar{\mathbf{f}}; s)$                       ▷ Revert speculation to form segmented scan
20: **procedure** UPDATEFIRSTSEGMENT($\mathbf{z}, \mathbf{f}; s$)
21:     $\bar{\mathbf{f}} \leftarrow$ MATMUL($\mathbf{f}, \mathbf{U}_s$)
22:     $\bar{\mathbf{f}}_{start} \leftarrow$ FILLS $(\bar{\mathbf{f}}; s)$                        ▷ $\bar{\mathbf{f}}_{start}(is : (i+1)s) = \bar{\mathbf{f}}(is)$
23:     $\mathbf{m} \leftarrow$ ISZERO (SUB $(\bar{\mathbf{f}}, \bar{\mathbf{f}}_{start})$)       ▷ $\mathbf{m}(i) = 1$ if $\bar{\mathbf{f}}(i)$ is equal to the first element of its segment
24:     $\mathbf{m} \leftarrow$ AND $(\mathbf{m}, \neg \mathbf{f})$                       ▷ No update on first element of each segment
25:     $\mathbf{w} \leftarrow$ FILLS $(\mathbf{z}; s)$                       ▷ $\mathbf{w}(is : (i+1)s) = \mathbf{z}(is)$ (unfiltered correction vector)
26:     $\mathbf{w} \leftarrow$ MASK $(\mathbf{w}, \mathbf{m})$                       ▷ Filtered (masked) correction vector
27:     $\mathbf{z} \leftarrow$ ADD $(\mathbf{z}, \mathbf{w})$

---

In particular, at each recursive step, the algorithm computes the "local" segmented scan of each $s$-block of the input $\mathbf{x}$ in two phases (**BLOCKSEGSCAN** in Algorithm A.2). In the first phase, we compute the *unsegmented* scans of each block by speculative matrix multiplications using $\mathbf{U}_s$ as the right matrix operand, see Line 17 of Algorithm A.2. In the second phase, the algorithm reverts any misspeculated segment using vector operations (REVSPEC in Algorithm A.2). Given an unsegmented scan of any $s$-block, Lemma A.2 describes an AC[0] circuit for REVSPEC, which

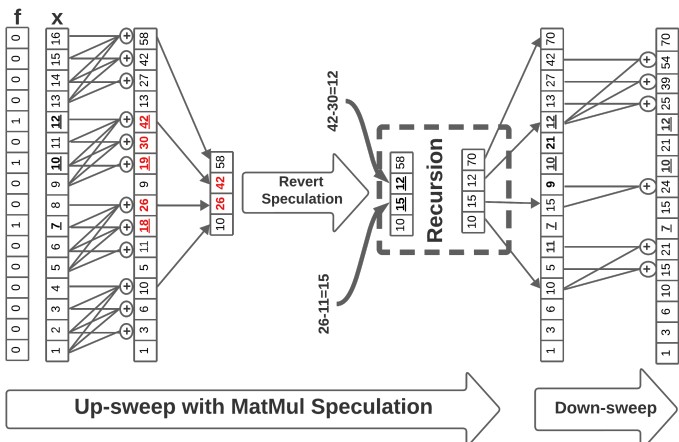

Figure A.1: Example of Algorithm A.2 with input of size 16 and $s = 4$. Red-coloured entries are computed speculatively and must be reverted before recursion. Underscored entries mark a segment start.

reverts the misspeculated scans. We highlight that such a circuit is only achievable because we have already precomputed the unsegmented scans of $\mathbf{x}$ and $\mathbf{f}$.

**Lemma A.2.** *Let $\mathbf{x}$ and $\mathbf{f}$ be a value and a boolean vector, each of size $s$. Assume that we have black-box access to a full scan of $\mathbf{x}$ and $\mathbf{f}$, i.e., $\widehat{\mathbf{x}}^\top = \mathbf{x}^\top \mathbf{U}_s$ and $\widehat{\mathbf{f}}^\top = \mathbf{f}^\top \mathbf{U}_s$, and assume that the elements of $\widehat{\mathbf{x}}$ and $\widehat{\mathbf{f}}$ fit in $B$-bit integers. Let $\mathbf{z}^*$ be the segmented scan of $\mathbf{x}$ w.r.t. $\mathbf{f}$. There exists an AC[0] circuit of size $\mathcal{O}(s^2 B + B^2)$ which, given $\widehat{\mathbf{x}}$ and $\widehat{\mathbf{f}}$, computes $\mathbf{z}^*$.*

*Proof.* We are given as input the two scanned vectors $\widehat{\mathbf{x}}$ and $\widehat{\mathbf{f}}$, both of size $s$. Recall the elements of $\widehat{\mathbf{f}}$ indicate the index of the segment where each element of $\mathbf{x}$ belongs. From each segment, we need to subtract the last element of the previous segment. In the final vector $\mathbf{z}^*$, the element $\mathbf{z}^*(j), j \in [s]$, is equal to the corresponding element $\widehat{\mathbf{x}}(j)$ minus the last element of the previous segment. More specifically, for all $j = 1, \ldots, s$, we must subtract from $\widehat{\mathbf{x}}(j)$ the element $\widehat{\mathbf{x}}(i)$, for some $i < j$, if and only if both the following conditions hold simultaneously:

1. $\widehat{\mathbf{f}}(i) \neq \widehat{\mathbf{f}}(i+1)$:   true, if segment ends at $i$,

2. $\widehat{\mathbf{f}}(j) = \widehat{\mathbf{f}}(i+1)$:   true, if j is in next segment.

To perform this subtraction efficiently, we will prepare an "update vector" $\mathbf{u}$, which contains $B$-bit integer elements, and each element $\mathbf{u}(j)$ will be subtracted from $\widehat{\mathbf{x}}(j)$. To that end, we first construct a binary "filter matrix" $\mathbf{Q}$, whose elements encode the above conditions 1. and 2., in particular:

$$\mathbf{Q}(i,j) = \begin{cases} \left(\widehat{\mathbf{f}}(i) \neq \widehat{\mathbf{f}}(i+1)\right) \wedge \left(\widehat{\mathbf{f}}(j) = \widehat{\mathbf{f}}(i+1)\right), \text{if } j > i \\ 0, \quad \text{otherwise.} \end{cases}$$

The elements of the matrix $\mathbf{Q}$ can be prepared by a circuit of constant depth and size $O(s^2 B)$.

Given $\mathbf{Q}$, we can write the elements of $\mathbf{u}$ as

$$\mathbf{u}(j) \leftarrow \bigvee_{i=1}^{j-1} \widehat{\mathbf{x}}(i) \wedge \mathbf{Q}(i,j),$$

where in this case we slightly abuse the $\bigvee$ and $\wedge$ notation, since $\widehat{\mathbf{x}}(i)$ is a $B$-bit integer, and therefore $\widehat{\mathbf{x}}(i) \wedge \mathbf{Q}(i,j)$ indicates that the "$\wedge$" gate operates on all the bits of $\widehat{\mathbf{x}}$ one-by-one. The size of this circuit is $\mathcal{O}(s^2 B)$, and the depth is again constant.

Finally, having prepared $\mathbf{u}$, we can set $\mathbf{z}^* \leftarrow \widehat{\mathbf{x}} - \mathbf{u}$. The final subtraction can be done in constant depth and size $\mathcal{O}(B^2)$. Putting everything together, the final circuit for $\mathbf{z}^*$ has constant depth and size $\mathcal{O}(s^2 B + B^2)$. $\qquad\square$

To use Lemma A.2 in Algorithm A.2, recall that the speculation prepares the entire vectors $\overline{\mathbf{x}} \leftarrow \text{MATMUL}(\mathbf{x}, \mathbf{U}_s)$ and $\overline{\mathbf{f}} \leftarrow \text{MATMUL}(\mathbf{f}, \mathbf{U}_s)$, which contain $\mathcal{O}(n/s)$ blocks of size $s$. By concatenating $\lceil n/s \rceil$ copies of the circuit of Lemma A.2, we can build a uniform family of circuits that implements the desired vector instruction REVSPEC. The total circuit size is $\mathcal{O}\left(\frac{n}{s}(s^2 B + B^2)\right) = \mathcal{O}\left(n(sB + B^2/s)\right)$, and the depth remains constant after concatenation.

In the last phase of Algorithm A.2, following the recursive call, the VCU are invoked to scalar-broadcast-vector-add the previous block's last entry (scalar) to the current block's first segment (UPDATEFIRSTSEGMENT in Algorithm A.2).

Indeed, UPDATEFIRSTSEGMENT corrects the values of the first segment of each $s$-block following the recursive call. Specifically, the last entry of a block must be broadcast added to the first segment of the corresponding next block. Again, the selective broadcast add is implemented with an unsegmented scan on $\mathbf{f}_s$ and (masked) vector operations.

Figure A.1 depicts an example of an execution of Algorithm A.2 for an input of size 16 and block size $s = 4$. In this example, matrix multiplication computes the unsegmented scans, and the misspeculated entries are coloured red. The reversion of the speculation is followed before the recursions take place. In the final step, vector-masked operations are employed to propagate the last entry of each block to the first segment of the next block.

**Remark A.1.** *The circuit of Lemma A.2 is ideal for our theoretical analysis, but it is rather complicated compared to the rest of the vector instructions in Table A.2. In a realistic setting, it would need to be implemented as a dedicated circuit inside the VCU, which can be prohibitive in terms of area. In Section A.3.6 we describe an alternative, Algorithm A.4, which simulates the REVSPEC instruction using only a few standard vector instructions, which are typically implemented in existing architectures.*

***Why is speculation used for segmented scan?*** At this point, we explain why speculation is employed and why it seems essential to compute segmented scans in the MMV-RAM model. The first argument is due to the linear algebraic structure of matrix multiplication, which does not allow us to handle arbitrary segment ranges with a single matrix multiplication operation. To see this, assume that we want to compute the segmented scans of $s$ consecutive blocks of length $s$ of some vector $\mathbf{x}$ of length $s^2$. Let $\mathbf{A}$ be a matrix with $s$ rows and columns containing $\mathbf{x}$ in row-wise order. Since the rows of $\mathbf{A}$ can have completely different segment ranges, each row of $\mathbf{A}$ must be multiplied with a *different* square matrix of size $s$ from the right to compute its local segmented scan. In the general case, it is impossible to encode all segmented scans of all rows with a single (right) matrix multiplication. Speculation overcomes this algebraic restriction by (speculatively) computing all $s$ unsegmented scans of each $s$-block via $\mathbf{A}\mathbf{U}_s$, which are later corrected using the VCU. The second reason why speculation is crucial, is that the segmented scans **cannot** be computed efficiently without the speculative scans, as already reported in Theorem 4.3.

**Reverting speculative scans using (masked) vector operations.** Here, we demonstrate that the REVSPEC vector operation can be simulated using multiple masked vector operations. We present two such algorithms.

First, Algorithm A.3 is simple and provides insight about the vector computations required. Its main drawback is that it performs reductions of $\mathcal{O}(s)$ numbers for each segment, hence requiring $\mathcal{O}(\log(s))$ steps and $\mathcal{O}(ns)$ work. To further reduce the work overhead of the speculation, we present Algorithm A.4, which has $\mathcal{O}(n \log(s))$ work and $\log(s) + \mathcal{O}(1)$ steps. The key difference of the two algorithms is that the latter uses a Kogge–Stone prefix masked reduction to avoid the work overhead (recomputation) compared to Algorithm A.3.

### A.3.7 PROOF OF THEOREM 4.3 (SEGMENTED SCAN)

Building on the analysis of Appendix A.3.6, we can now prove Theorem 4.3.

*Proof.* The analysis is almost the same as in Theorem 4.1. The recursion depth is again $\mathcal{O}(\log_s(n))$. Regarding the total work, in each recursive step $t = 1, \ldots, \lfloor \log_s(n) \rfloor$ we need to execute four MATMUL operations of length $\mathcal{O}(n/s^t)$, and few vector instructions. The most expensive VCU instruction is REVSPEC, which, from Table A.2, requires $\mathcal{O}(\frac{n}{s^t}(sB + B^2/s))$. Thus, the total work

---

**Algorithm A.3** Revert Speculative Prefix Sums

1: **procedure** REVERTSPECSIMPLE($\mathbf{z}, \mathbf{x}, \mathbf{f}, s$)        ▷ Works in-place on $\mathbf{z}$
2:     $\overline{\mathbf{f}} \leftarrow \text{MATMUL}(\mathbf{f}, \mathbf{U}_s)$
3:     **parfor** each $s$-block $(\mathbf{z}_s, \mathbf{x}_s, \overline{\mathbf{f}}_s)$ **do**
4:        $\mu \leftarrow \overline{\mathbf{f}}_s(s-1)$
5:        **if** $\mu == 0$ or ($\mu == 1$ and $\overline{\mathbf{f}}_s(0) == 1$) **then**
6:           **return**        ▷ Speculation is correct.
7:        **if** $\mu == s$ **then**
8:           $\mathbf{z}_s \leftarrow \mathbf{x}_s$        ▷ $s$ segments. Revert.
9:           **return**
10:      **parfor** idx $= \overline{\mathbf{f}}_s(0) + 1, \ldots, \mu$ **do**
11:         offset $\leftarrow \text{SUM}(\text{LESSTHAN}(\overline{\mathbf{f}}_s, \text{idx})) - 1$
12:         $\mathbf{z}_s \leftarrow \mathbf{z}_s - (\overline{\mathbf{f}}_s == \text{idx}) \cdot \mathbf{z}_s(\text{offset})$

---

**Algorithm A.4** Revert Speculative Scans with MMV-RAM Vector instructions

1: **procedure** REVERTSPECVECTOR($\mathbf{z}, \mathbf{x}, \mathbf{f}, s$)        ▷ Works in-place on $\mathbf{z}$
2:     $\overline{\mathbf{f}} \leftarrow \text{MATMUL}(\mathbf{f}, \mathbf{U}_s)$
3:     **parfor** each $s$-block $(\mathbf{z}_s, \mathbf{x}_s, \overline{\mathbf{f}}_s)$ **do**
4:        $\mu \leftarrow \overline{\mathbf{f}}_s(s-1)$
5:        **if** $\mu == 0$ or ($\mu == 1$ and $\overline{\mathbf{f}}_s(0) == 1$) **then**
6:           **return**        ▷ Speculation is correct.
7:        **if** $\mu == s$ **then**
8:           $\mathbf{z}_s \leftarrow \mathbf{x}_s$        ▷ $s$ segments. Revert.
9:           **return**
10:      $\mathbf{q}_s \leftarrow \text{XOR}\left(\text{SHIFTL}\left(\overline{\mathbf{f}}, 1\right), \mathbf{f}\right)$
11:      $\mathbf{m}_s \leftarrow \text{SHIFTR}(\mathbf{q}_s, 1)$
12:      $\mathbf{w}_s \leftarrow \text{SHIFTR}\left(\text{MASK}\left(\mathbf{z}_s, \mathbf{q}_s\right), 1\right)$        ▷ Correction vector.
13:      **for** $j = 0, \ldots, \lceil \log(s) \rceil$ **do**
14:         $b \leftarrow 2^j$
15:         $\overline{\mathbf{m}}_s \leftarrow \text{SHIFTR}\left(\mathbf{m}_s, b\right)$
16:         $\overline{\mathbf{w}}_s \leftarrow \text{SHIFTR}\left(\mathbf{w}_s, b\right)$
17:         $filter_s \leftarrow \text{LEQ}\left(\text{ADD}\left(\mathbf{m}_s, \overline{\mathbf{m}}_s\right), 2\right)$
18:         $\mathbf{m}_s \leftarrow \text{MASK}\left(\text{ADD}\left(\mathbf{m}_s, \overline{\mathbf{m}}_s\right), filter_s\right)$
19:         $\mathbf{w}_s \leftarrow \text{MASK}\left(\text{ADD}\left(\mathbf{w}_s, \overline{\mathbf{w}}_s\right), filter_s\right)$
20:      $\mathbf{z}_s \leftarrow \text{SUB}\left(\mathbf{z}_s, \mathbf{w}_s\right)$

---

becomes:

$$W(t) = W(t+1) + 4\mathcal{M}\left(\tfrac{n}{s^t}\right) + \mathcal{O}\left(\tfrac{n}{s^{t-1}}\left(sB + \tfrac{B^2}{s}\right)\right)$$

$$= \mathcal{O}\left(\mathcal{M}(\tfrac{n}{s}) + n(sB + \tfrac{B^2}{s})\right).$$

Once again, in the "down sweep" phase, the two operations that can increase the magnitude of elements are two consecutive MATMUL operations. REVSPEC can only decrease the magnitude. In the "up-sweep" phase, we have an ADD instruction that can increase the magnitude of elements by a factor of two, or, equivalently, by a single bit. Therefore the number of required bits to avoid overflow is upper bounded by the bits required in Theorem 4.1.    □

### A.4 ANALYSIS OF APPLICATIONS OF SEGMENTED OPERATIONS

### A.4.1 INTEGER MULTIPLICATION, ELEMENT-WISE VECTOR MULTIPLICATION

Here we outline how to apply out segmented operations MMV-RAM analysis to other basic applications, starting with the multiplication of two integers.

**Lemma A.3.** *Let $a$ and $b$ be two integers with magnitudes $|a|, |b| \leq M \leq 2^B$. In the MMV-RAM model, we can compute the product $ab$ with:*

- $\mathcal{O}(\log_s(B))$ *steps and* $\mathcal{O}\left(\mathcal{M}(\tfrac{B}{s}) + sB^2\right)$ *work,*

- $B \in \mathcal{O}(\log(M))$ *bits-per-element to avoid overflow.*

*Proof.* Let $m = \lceil \log(M) \rceil$. Following the "school method" (e.g. (Vollmer, 1999, Chapter 1)), we know that we can write $ab = \sum_{i=0}^{m-1} c_i$, where:

$$c_i := \begin{cases} \overbrace{0 \ldots 0}^{m-i-1} a_{m-1} \ldots a_1 a_0 \overbrace{0 \ldots 0}^{i}, & \text{if } b_i = 0, \\ \overbrace{0 \ldots 0}^{2m-1}, & \text{otherwise.} \end{cases}$$

Now, each $c_i$ can be prepared with a circuit of size $\mathcal{O}(m)$ in constant depth, which gives a total size of $\mathcal{O}(m^2)$ for all $c_i$'s, and each $c_i$ consists of $2m - 1$ bits. We can put the $c_i$'s in a vector $\mathbf{c}$ of length $B$, and use Theorem 4.1 to compute a full scan of $\mathbf{c}$ in $\mathcal{O}(\log_s(m))$ steps and $\mathcal{O}(\mathcal{M}(\frac{m}{s}) + sm^2)$ work. The last element of the scanned vector contains $ab$. In terms of avoiding overflow, Theorem 4.3 indicates that $B \in \mathcal{O}(\log(m) + m) = \mathcal{O}(\log(M))$ bits are sufficient. □

This directly extends to element-wise vector multiplication.

**Corollary A.1.** *Let $\mathbf{x}$ and $\mathbf{y}$ be two vectors with integer elements bounded by $|\mathbf{x}(i)|, |\mathbf{y}(i)| \leq M$. We can compute their element-wise product with an algorithm $\mathbf{z} \leftarrow \text{MULT}(\mathbf{x}, \mathbf{y})$ in the MMV-RAM model, with:*

- $\mathcal{O}(\log_s(B))$ *steps,* $\mathcal{O}\left(\mathcal{M}(\frac{nB}{s}) + nsB^2\right)$ *work.*

- $B \in \mathcal{O}(\log(nM))$ *bits to avoid overflow.*

*Proof.* For each element $\mathbf{z}(i) = \mathbf{x}(i)\mathbf{y}(i)$, we can compute a SCAN as in Lemma A.3. This can be done collectively for all $i$ with a segmented scan operation (Algorithm A.2), where each segment corresponds to one $\mathbf{z}(i)$. □

### A.4.2 MATRIX MULTIPLICATION

The aforementioned techniques can be used to compute the product of two matrices with integer entries. In this section, we provide a work/depth analysis of the standard, inner product-based matrix multiplication algorithm in the MMV-RAM model. The segmented scan Algorithm A.2 and the integer vector multiplication algorithm of Corollary A.1 form the main building blocks. The main idea here is to exploit the MMU to perform the final reductions, after the scalar (or block) multiplications. To see the potential impact, consider first a matrix multiplication algorithm that is split in two-phases: (i) scalar products and (ii) reductions to form the final matrix product. Even if we obtain the scalar multiplications (i) "for free", we still need to compute the reductions (ii). If we use only the VCU for this task, we need to execute $\Omega(\log(n)/\log\log(n))$ steps. The same holds if, instead of scalar products, we use the MMU in phase (i) to compute matrix products of small $s \times s$ blocks. That is, we still need to reduce these small products in the second phase, in order to form the final matrix product. The reductions require $\Omega(\log(\frac{n}{s})/\log\log(\frac{n}{s}))$ steps with the VCU. Notably, in the following Theorem A.2 we obtain an algorithm with only $\tilde{\mathcal{O}}(\log_s(nB))$ step complexity and nearly $\mathcal{O}(n^3)$ work.

**Theorem A.2.** *Let $\mathbf{A}$ and $\mathbf{B}$ be two $n \times n$ matrices with non-negative integer entries that are bounded in magnitude by $M$. In the MMV-RAM model, we can compute the product $\mathbf{C} = \mathbf{AB}$ with*

- $\mathcal{O}(\log_s(nB))$ *steps,*

- $\mathcal{O}\left(\mathcal{M}(\frac{n^3 B}{s}) + n^3 s B^2\right)$ *work,*

- $\mathcal{O}(\log(nM))$ *bits-per-element to avoid overflow.*

*Any matrix multiplication algorithm that requires to perform reductions between blocks of size $\mathcal{O}(s) \times \mathcal{O}(s)$ with the VCU requires $\Omega\left(\frac{\log(n/s)}{\log(\log(n/s))}\right)$ steps.*

*Proof.* There exists an AC[0] circuit with size $O(n^3 B)$ which prepares two vectors $\mathbf{a}$ and $\mathbf{b}$, with size $n^3$ each. In particular, $\mathbf{a}$ contains $n$ concatenated copies of a flattened version of the matrix $\mathbf{A}$, while $\mathbf{b}$ contains $n$ copies of the first column of $\mathbf{B}$, followed by $n$ copies of the second column, $n$ copies of the third column, and so on. This way, the vector multiplication $\mathbf{c} \leftarrow \text{MULT}(\mathbf{a}, \mathbf{b})$, will contain all the integer scalar products needed by the standard dot product-based matrix multiplication algorithm. Using Corollary A.1 the total work is $\mathcal{O}\left(\mathcal{M}(\frac{n^3 B}{s}) + n^3 s B^2\right)$ and the number of steps $\mathcal{O}(\log_s(B))$. We then do an additional segmented scan using Algorithm A.2, which, from Theorem 4.3, requires $\mathcal{O}\left(\mathcal{M}(\frac{n^3}{s}) + nB(s + \frac{B}{s})\right)$ work and $\mathcal{O}(\log_s(n))$ steps. In terms of bits to avoid overflow, the first segmented scan requires $\mathcal{O}(\log(M))$ bits from Lemma A.3, while the second requires $\mathcal{O}(\log(n) + \log(M))$, from Theorem 4.3. $\qquad\square$

### A.4.3 CSR SPMV

We finally provide the analysis of our SpMV algorithm, which uses segmented sums (SCD) as a subroutine.

**Theorem A.3** (SpMV). *Let $\mathbf{A}$ be an $n \times n$ sparse matrix in CSR format and $\mathbf{x}$ be an input vector, both with non-negative integer entries that are bounded in magnitude by $M$. In MMV-RAM, we can compute the product $\mathbf{y} = \mathbf{A}\mathbf{x}$ with*

- *$\mathcal{O}(\log_s(nB))$ steps, $\mathcal{O}\left(\mathcal{M}(\frac{nB}{s}) + n^2 B + nsB^2\right)$ work,*

- *$\mathcal{O}(\log(nM))$ bits-per-element to avoid overflow.*

*Any MMV-RAM algorithm that uses only the VCU requires $\Omega\left(\frac{\log(n)}{\log(\log(n))}\right)$ steps.*

*Proof.* The first step of Algorithm 4.3 requires to gather the elements of $\mathbf{x}$ in a vector, indexed by the values of the column vector of $\mathbf{A}$. This can be achieved with an AC[0] circuit similar to the one in Appendix A.3.5, with depth one and size $\mathcal{O}(n^2 B)$. In the next step, we use Corollary A.1 to multiply the gathered values with the value vector of $\mathbf{A}$, using $\mathcal{O}(\log_s(B))$ steps, and $\mathcal{O}\left(\mathcal{M}(\frac{nB}{s}) + nsB^2\right)$ work. Finally, we use SCD, which, from Theorem 4.2, requires $\mathcal{O}(\log_s(n))$ steps and $\mathcal{O}(\mathcal{M}(\frac{n}{s}) + nB(n + B))$ work.

This gives a total of $\mathcal{O}(\log_s(n) + \log_s(B)) = \mathcal{O}(\log_s(nB))$ steps and $\mathcal{O}\left(\mathcal{M}(\frac{nB}{s}) + nsB^2 + n^2 B\right)$ work. $\qquad\square$

### A.5 ADDITIONAL DETAILS ON IMPLEMENTATIONS, DATASETS, AND EXPERIMENTS

### A.5.1 SPMV IMPLEMENTATION

Here, we provide more details about the SpMV algorithm. In Algorithm A.5, we list the variant of Algorithm 4.3 that is implemented and used in our experimental evaluation and, most importantly, takes into account the corner case where a sparse matrix has empty rows; see (Liu & Vinter, 2015b;a).

---

**Algorithm A.5** MMV-RAM Sparse CSR Matrix-Vector (SpMV)

---
1: **procedure** MMVRAMCSRSPMV($\mathbf{A}, \mathbf{x}$) $\qquad\qquad\qquad\qquad\qquad \triangleright$ $\mathbf{A}$ is $m \times n$ CSR matrix
2: $\qquad \mathbf{z} \leftarrow$ CSRGATHER($\mathbf{A}.\mathbf{val}, \mathbf{A}.\mathbf{col}, \mathbf{x}$)
3: $\qquad \hat{\mathbf{z}} \leftarrow$ SCAN($\mathbf{z}$)
4: $\qquad \mathbf{w} \leftarrow$ GATHERSPMV($\hat{\mathbf{z}}, \mathbf{A}.\mathbf{row}$)
5: $\qquad$ **return** DIFF($\mathbf{w}$)

---

Algorithm A.6 describes the CSRGATHER step of our SpMV algorithm. Given $\mathbf{A}$ and $\mathbf{x}$ the algorithm compute $\mathbf{z}$ such that:

$$\mathbf{z}(i) := \mathbf{val}(i) * \mathbf{x}(\mathbf{col}(i)) \quad 0 \le i < \text{nnz}. \tag{4}$$

In our implementation, the algorithm collects the whole vector $\mathbf{x}$ into the scratchpad memory of each AIVs and then performs a gather and an element-wise multiplication. A current limitation of

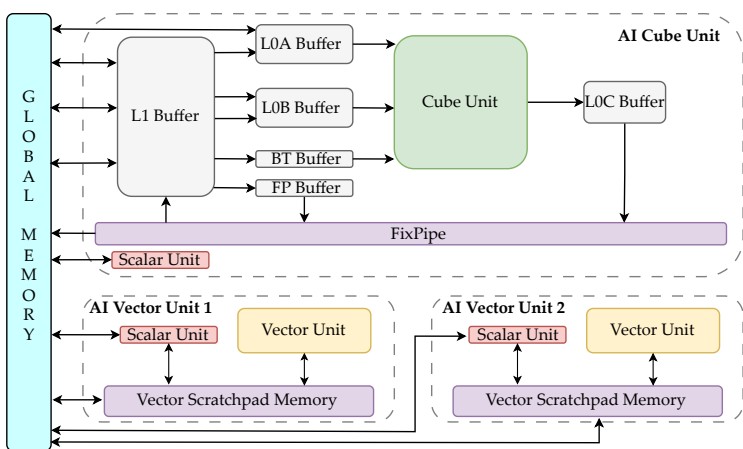

Figure A.2: Architecture of an AI-core of the Ascend 910B AI accelerator. Each AI-core consists of one Cube (matrix multiplication) unit and two vector cores.

the algorithm is that it only works for input vectors $\mathbf{x}$ of length up to 70K since the whole vector $\mathbf{x}$ must be collected on each AIV core on the gather step (Line 4). Finally, the GATHERSPMV

---

**Algorithm A.6** CSR Gather step of (Blelloch et al., 1993), see also (Liu & Vinter, 2015b).

---

1: **procedure** CSRGATHER($\mathbf{val}, \mathbf{col}, \mathbf{x}$)
2:     $\mathbf{z} \leftarrow$ Zeros vector of length nnz
3:     **for** each $s$-tile ($\mathbf{val}_s, \mathbf{col}_s, \mathbf{z}_s$) **do**
4:         $\mathbf{w}_s \leftarrow$ Gather($\mathbf{x}, \mathbf{col}_s$)
5:         $\mathbf{z}_s \leftarrow$ Mul($\mathbf{val}_s, \mathbf{w}_s$)                               ▷ Element-wise.
6:     **return** $\mathbf{z}$

---

Algorithm A.7 implements a specialized step to collect the last element of each row after the SCAN step and add zeros in correspondence of the initial matrix empty rows.

---

**Algorithm A.7** Specialized gather for SpMV.

---

1: **procedure** GATHERSPMV($\mathbf{y}, \mathbf{row}$)                               ▷ len($\mathbf{y}$) = nnz
2:     $s \leftarrow$ tiling parameter based on nnz and $m$.
3:     ($\mathbf{z}$, nnz) $\leftarrow$ HANDLELEADINGZEROS($\mathbf{y}, \mathbf{row}$)
4:     $\mathbf{z}$(nnz :) $\leftarrow$ GATHER($\mathbf{y}, \mathbf{row}, s$, offset = 1)
5:     **return** $\mathbf{z}$
6: **procedure** HANDLELEADINGZEROS($\mathbf{y}, \mathbf{row}$)
7:     Let $\mathbf{z} \leftarrow \mathbf{y}, i \leftarrow 0$
8:     **while** $\mathbf{row}(i) == 0$ **do**
9:         $\mathbf{z}(i) = 0$
10:         $i \leftarrow i + 1$
11:     **return** $\mathbf{z}, i$                               ▷ $i$: # of leading zero rows of $\mathbf{A}$.

---

A.5.2    DETAILS ON ASCENDC DEVELOPMENT

The programming model is asynchronous and pipeline-based, as multiple AI-core executions can be pipelined while data flows from one unit to the other, optimizing performance. Computational kernels could be written in AscendC, a programming language built on top of C++ allowing for fine-grained control over the hardware components of Ascend. As the accelerator specifically targets AI workloads, in which we can usually apply quantization to handle data sizes, it mainly offers support for int8, float16, and float32, making the accelerator suitable for both training and inference steps; see (Liao et al., 2021) for further details.

**Implementation of Ascend operators.**   Here, we provide implementation details regarding the parallel primitive operators. For unsegmented scans, we use the best available scan implementation

for Ascend, which also makes heavy use of the MMU. This scan implementation, SCAN, supports both as inputs half/float16 and integers with bit-width 8 (int8) with output data types float32 and int32, respectively. For vector differentiation (DIFF), we use the implementation provided by Ascend's software stack. For compress (COMPRESS), we use the best available implementation that also makes heavy use of the MMU. COMPRESS is implemented by first scanning $\mathbf{f}$, which has data type int8, followed by a gather vector operation. Finally, we implemented the REVERT kernel as a vector-only operator, as it is not straightforward how to use the MMU for this task.

### A.5.3 DATASET DETAILS

Tables A.3 and A.4 show the matrix characteristics of the matrices used in Section 5.

Table A.3: SuiteSparse matrices from Figure 5.1.

| Name | Dimensions | $nnz$ | $nnz$ per row | | |
| | | | min | avg | max |
|---|---|---|---|---|---|
| Protein | $36K \times 36K$ | 4.3M | 2K | 2K | 2K |
| Epidemiology | $526K \times 526K$ | 2.1M | 18 | 119 | 204 |
| Economics | $207K \times 207K$ | 1.3M | 1 | 6 | 44 |
| Circuit5M | $5.6M \times 5.6M$ | 2.1M | 18 | 119 | 204 |
| ASIC_680K | $683K \times 683K$ | 3.9M | 1 | 6 | 395K |

Table A.4: SuiteSparse atrices from Fig.5.2, with sizes ranging between $30K \times 30K$ and $70K \times 70K$.

| Name | $nnz$ | ID | Kind |
|---|---|---|---|
| enron | $276K$ | 2444 | Directed Graph |
| me2010 | $336K$ | 2602 | Undirected Weighted Graph |
| struct3 | $1.17M$ | 807 | Structural Problem |
| water_tank | $2.04M$ | 1880 | Comput. Fluid Dynamics |
| srb1 | $2.96M$ | 1282 | Structural Problem |
| cant | $4.01M$ | 2375 | 2D/3D Problem |
| Chebyshev4 | $5.38M$ | 1867 | Structural Problem |
| TSOPF_FS | $8.77M$ | 2225 | Power Network Problem |
| mip1 | $10.35M$ | 1385 | Optimization Problem |
| crankseg_1 | $10.61M$ | 1257 | Structural Problem |

### A.5.4 ADDITIONAL EXPERIMENTS

In this Appendix we provide some additional experimental results and insights of the main algorithms. We evaluate our multi-AI-core implementation of the segmented sum (SCD) and segmented scan (SSCR) algorithms, presented in Section 4. The objective here is to provide an analysis of the relative performance of the individual parallel primitives used in these algorithms. Such a "roof-line" type of analysis can reveal opportunities for performance improvements, that can lead to highly-optimized future implementations. By default, in the experiments that follow the measurements are averaged (arithmetic mean) over 100 repetitions, unless explicitly stated otherwise. Figure A.3 depicts the performance (bandwidth) of the single-core segmented scan algorithm on various segment density levels, in the range of $0.01\%$ up to $1\%$, drawn uniformly at random. Density is the ratio of the number of ones in $\mathbf{f}$ over the length of $\mathbf{f}$. For density equal to $0.01\%$, the algorithm performs equally with the unsegmented case, which serves as our baseline, even marginally better due to the particular double-vector core implementation that we used. For density $\approx 0.1\%$, it reaches nearly $90\%$ of the unsegmented performance. The performance decreases for densities $> 0.3\%$, but in all cases it remains within $\gtrsim 20\%$ of the unsegmented scan performance.

Figure A.4 (left) shows the bandwidth performance of each parallel primitive (SCAN, COMPRESS, DIFF and REVERT), compared to a memcopy operator, for increasing input lengths. The plot demonstrates that the performance of all operators, which are memory-bound, is close to the peak memory bandwidth offered by the device (800GB/s). The bandwidth performance of REVERT for segment density 0.001 is surprisingly higher than the performance of our best SCAN operator, but it significantly degrades for density 0.01. COMPRESS and REVERT (0.001) achieve the lowest bandwidth overall, which is expected due to the irregular memory transfers that then need to execute.

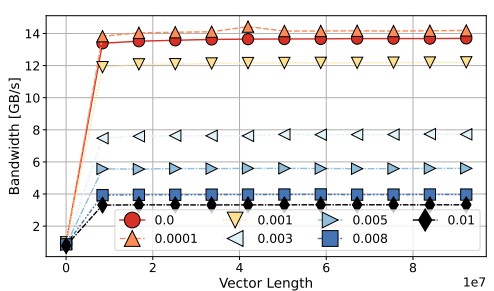 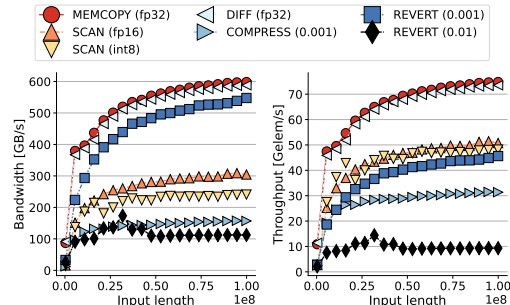

Figure A.3: Single AI-core segmented scan ($s = 128$) over uniformly random segments with varying densities (zero-density is the unsegmented case, i.e., the baseline).

Figure A.4: Bandwidth/throughput of parallel primitives vs. memcopy (peak theoretical bandwidth is 800GB/s).

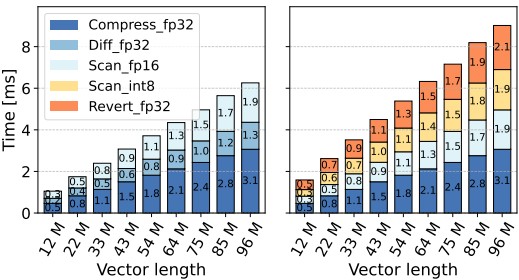

Figure A.5: Execution time breakdowns for SCD (Alg. 4.1, left) and SSCR (Alg. 4.2, right), for varying lengths.

On the right of Figure A.4, we plot the performance of the operators with respect to the number of elements of the input vector $\mathbf{x}$ that are processed per second. The $y$-axis depicts billions (Giga) elements per second (Gelem/s). The performance (Gelem/s) of the scan operators (both fp16 and int8) is better compared to the COMPRESS and REVERT operators, which is expected since both of them read additional int32 inputs to encode the segments. The performance of REVERT with segment density equal to 1% has the lowest performance around 10 Gelem/s.

Figure A.5 illustrates an execution time breakdown of the parallel primitives used in the SCD (left figure) and SSCR (right figure) algorithms. In both figures, the matrix multiplication tile size, $s$, is set to 128 and the density of segments is 0.1%. Both figures show that COMPRESS takes a large fraction of the execution time, i.e., around 50% and around 33% for SCD and SSCR, respectively. The DIFF operator implementation on Ascend has almost identical performance to the data copy operator. Both COMPRESS and SCAN have similar performance in the left figure. In the SSCR case, the REVERT operation takes a considerable fraction of the elapsed time, but COMPRESS remains the bottleneck.

