# OpenReview forum: "Segmented Operations using Matrix Multiplications"
_ICLR.cc/2026/Conference — Submitted to ICLR 2026_

### Official Review · Reviewer_U87B · 2025-10-24

**Soundness:** 3
**Presentation:** 4
**Contribution:** 3
**Rating:** 4
**Confidence:** 2

**Summary:**

Introduces a new compute model called MMV-RAM, consisting scalar, vector & matrix units with memory. This extends the existing Vector-RAM model with a matrix unit. The vector and matrix units are distinguished by their circuit complexity class - the vector unit operations must be in AC[0] (unbounded fan-in, polynomial width, constant depth), while the matrix unit is restricted to linear transforms, but these are not in AC[0].

The work describes parallel algorithms for scan, segmented scan and segmented sum, with good computational complexity under this model. These can be used to support a sparse matrix-vector product (SpMV). Finally, a demonstration in software on deep learning hardware shows the benefit of these methods that use the matrix unit for segmented scan and SpMV.

**Strengths:**

The paper is well-written and the claims and results are clear. The question of which compute model to use for algorithm analysis is relevant and interesting to the community. Particular strengths:

 - Clarity of notation, examples and algorithms in general, as well as diagrams and plots.
 - Simplicity of the proposed model, MMV-RAM.
 - Section 4.2 on applications of segmented operations was helpful for understanding the impact of the main results.
 - Details of proofs, algorithms, and experimental datasets in the appendix are comprehensive and appreciated.
 - The practical algorithm benchmarks in software are appreciated.

**Weaknesses:**

I am not experienced in circuit complexity theory, and so I hope to be corrected on my main concern, regarding the bounds that are shown to justify the matrix unit. In Theorems 4.1 and 4.2, the step bound is given $\mathcal{O}(\log_s(n))$ with matrix unit, and $\Omega(\frac{\log(n)}{\log(\log(n))})$ without. Since $s$ is a model parameter that does not depend on $n$ (L184), these bounds seem to me to overlap, as if I treat $s$ as a constant, $\mathcal{O}(\log_s(n)) = \mathcal{O}(\frac{\log(n)}{\log(s)}) = \mathcal{O}(\log(n)) \supset \mathcal{O}(\frac{\log(n)}{\log(\log(n))})$. So, for any given $s$, I can't see how these bounds would justify MMV-RAM over Vector-RAM?

Specific concerns:

 1. The experiments are interesting, but from my perspective do little to justify the cost model. Specifically, I imagine an accelerator with two vector units, one fast & one slow, could exhibit identical behaviour to the Ascend with vector & matrix units. To understand the importance of separating vector and matrix units in the cost model, it might be more convincing to look at the hardware implications: for an example from another domain, Rouhani et al. (2023) demonstrate their low-precision format ideas using ASIC synthesis and area estimation - this is likely too far, but for this work, some analysis that allows for more exploration of design parameters rather than being constrained to a specific accelerator might be more convincing.
 1. The model assumes computing an $n \times s$ with $s \times s$ product in a single step, which seems quite far from hardware designs. This is addressed in L802, Appendix A.2, which describes tiling an $s \times s$ square product, but since we are concerned with parallel algorithms, I wonder if would be simpler to specify the matrix unit as multiplying $s \times s$ and $s \times s$ in a single step, as we can execute arbitrarily many (e.g. $\lceil \frac{n}{s} \rceil$) of these operations in parallel, in any case.

Minor concerns:

 - The paper is somewhat lacking in breadth/depth of insights gleaned from the MMV model. For example, an explicit comparison of any advantage of MMV over TCU for the unsegmented scan of Zouzias and McColl (2023), as well as the introduced SCD and SSCR algorithms would be useful. Or, some deeper understanding of the resource requirements for balancing matrix and vector units, beyond the observation that both are necessary in Section 3.1.
 - I understand scan/segment-scan to typically become memory-bound when operands need to come from a HBM/DDR memory system, and that vector units can typically keep up with the memory system in this case. Is this the case for the Ascend AI 910B accelerator?
   - Aside: it would be useful to quote the L3 cache and scratchpad sizes referenced in L431.
   - I am surprised how slow the vector-only implementation runs in Figure 5.1, compared with CPU. What resource utilisation is this achieving versus peak vector arithmetic throughput?

---

_Darvish Rouhani, B., Zhao, R., Elango, V., Shafipour, R., Hall, M., Mesmakhosroshahi, M., More, A., Melnick, L., Golub, M., Varatkar, G. and Shao, L., 2023, June. With shared microexponents, a little shifting goes a long way. In Proceedings of the 50th Annual International Symposium on Computer Architecture (pp. 1-13)._

_Zouzias, A. and McColl, W.F., 2023, August. A parallel scan algorithm in the tensor core unit model. In European Conference on Parallel Processing (pp. 489-502). Cham: Springer Nature Switzerland._

**Questions:**

See questions mentioned above in "weaknesses". Many thanks!

---

> ### Author Response · Authors · 2025-11-19
>
> We want to thank the reviewer for the interest in our work and for the thorough feedback. Below, we provide responses to all the interesting questions that were raised.
>
> ### Main question
> - "In Theorems 4.1 and 4.2, the step bound is given..."
>
> **Response:** This is a really great question that was also asked by another reviewer. We understand that this part causes confusion, and the reviewers were right to raise this issue.
> In the final paper (if it gets accepted), we will expand the discussion and try to explain this part better.
>
> We start by highlighting a subtle part in the analysis of the reviewer: the complexity **upper bound** of the matmul-based algorithm is $O(\log(n)/\log(s))$, while for *any* vector-only algorithm we have a **lower bound** of $\Omega(\log(n)/\log(\log(n)))$. The latter does not imply that there is actually any algorithm that achieves this lower bound. This is a critical point. It might very well be the case that the best-known vector-only algorithm achieves $O(\log(n))$ depth, instead of $O(\log(n)/\log\log(n))$.
>
> But let us ignore this fact for now in order to move the discussion forward. Let us assume that there is indeed a vector-only algorithm with depth $O(\log(n)/\log\log(n))$. If $s$ is treated as a constant, indeed, there is no speedup **asymptotically** (but also no slowdown since MMV-RAM is a superset of Vector-RAM). However, $s$ is not a constant, it is a model parameter, and this drives the discussion towards what in the literature refers to as "parametrized complexity".
>
> It is not hard to calculate the "regimes" of $n$ and $s$ that provide a speedup. We can replace the $O(\cdot)$ and $\Omega(\cdot)$ with constants, and write $T_1=c_1\log(n)/\log(s)$ and $T_2=c_2\log(n)/\log(\log(n))$. Then we search for the $s$ that satisfies $T_2>T_1$, which gives $s>c_3\log(n)$. This can be interpreted also as follows: if $s$ is a fixed model parameter, we can expect to get a speedup for all $n<2^{s/c_3}$ (the constant $c_3$ can be calculated exactly but we skip this for brevity). Typical values for $s$ in current hardware and software libraries range around $32,64$ or $128$. Moreover, it is possible to efficiently simulate matmuls with even larger $s$, e.g. up to $1024$, depending on the hardware/software capabilities. If we set $c_3=1$ for simplicity, and $s=128$, then we have a provable **theoretical** speedup for all $n<2^{128}\approx 3.4\times 10^{38}$. This preliminary analysis for the range of $n$ is certainly encouraging for further investigation.
>
> In general, the analysis with respect to $s$ allows us to obtain **explicit trade-offs** between the MMU size and the achievable speedup that someone might hope for.
> Of course, a skeptical algorithm/hardware designer might ask:
> - Are these speedups significant?
>
> Arguing about the importance of such speedups requires domain specific considerations depending on the target use case. The reported lower/upper complexity bounds can help a domain expert to decide whether to use MMUs for a target application can be beneficial.
> For example, both of the following arguments are valid:
>
> - "The theoretical speedup is only logarithmic compared to $s$, it is not worth increasing the MMU size by a linear factor to get only a logarithmic speedup".
> - "We can actually get a speedup even for memory-bound kernels simply by increasing the MMU size, what a blessing!"
>
> In conclusion, our results can be interpreted both as positive or as negative, depending on the application. We certainly hope that the AI accelerator industry will carefully take into account these types of theoretical results (also from other related works, not only ours) in order to make informed decisions on the hardware components of their chips (specifically the matmul / vector units), before investing heavily on manufacturing.
>
> Thanks again for mentioning this unclear part in our paper. We are more than happy to chat more during the discussion period (or at the conference), if there are further questions!

---

> > ### Author Response · Authors · 2025-11-19
> >
> > ### Specific concerns
> > 1. "The experiments are interesting, but from my perspective."
> >
> > **Response:**
> > This question seems to have several parts, we will try to answer it as best as possible. Regarding the design/choice of a cost model for algorithm analysis, it is well-known from the long history and rich literature on models of computation that there is no unique model "to rule them all". Different models have been proposed for decades, both from the theory and the engineering side, and every model is strong in some aspects and weaker in others.
> >
> > This work has a very clear target: we want to do work-depth analysis, in the classic sense such as in the celebrated Vector-RAM model, one of the most important models for shared-memory parallel algorithms. As you can observe from our analysis, it required a very big effort from the theory side to incorporate all the needs of modern hardware that includes not only vector cores, but also these new types of matrix multiplication units.
> > The reason why we decided to do this theory is that the pre-existing models did not allow us to do a proper work-depth algorithm analysis, at least not to a satisfactory level.
> >
> > Now, the reviewer makes a great point (intentionally or not): do we need matmul units, or can we replace it with a "stronger vector unit" instead? If you follow our theoretical analysis of MMV-RAM, you will see that there are many delicate questions to answer in the design:
> > - What kind of instructions can the "stronger vector unit" implement?
> > - What is the associated cost for each instruction?
> > - If we have this unit, do we even need the weaker vector unit at all? Do they complement each other (in the sense of our Theorem 3.1), or is one of them deprecated?
> > - How does the work-depth analysis look in the "weak/strong vector" model?
> > - What kind of algorithms can benefit from implementations in the new model?
> >
> > We carefully answered all of these questions and more in MMV-RAM. Our choice to incorporate specifically a vector and a matmul unit is due to the emerging hardware architectures, which include such units. Interestingly, we were actually surprised ourselves when we arrived to the main results: initially, we thought that our final verdict would be that "matmul units are useless theoretically, we can do everything with vector cores". In Theorem 3.1 we disproved ourselves. With careful analysis, both of them are necessary to get the best possible algorithms (at least from the theory side).
> >
> >
> > In the final part of the question, the reviewer mentions that "some analysis that allows for more exploration of design parameters rather than being constrained to a specific accelerator might be more convincing". We do not fully understand this part, but let us highlight that MMV-RAM is not constrained to any specific accelerator. The only assumption is that you have three units: a scalar, a vector, and a matmul unit. At the same time, we want to keep the model simple and usable. By adding more parameters to any model, it certainly becomes more realistic, but it also starts becoming less usable, because the complexity increases significantly. This is a known issue in the "models of computation" literature. We are not sure if this answers this last part but we can discuss further if more details are provided.
> >
> > 2. "The model assumes computing..."
> >
> > **Response:** We think that this is already covered in Appendix A.1 "EXTENSIONS OF MMV-RAM", and in particular in Appendix A.1.2  "EXTENDING THE RIGHT-HAND-SIDE OF THE MMU". There, we explicitly mention what the reviewer specifically mentions. From our analysis in that appendix, there seems to be no benefit if we allow multiple independent $s\times s$ matmuls instead of a large $n\times s \times s$ matmul. This seems counter-intuitive initially, but the complexity bounds seem pretty tight. We are happy to discuss this further.

---

> > > ### Author Response · Authors · 2025-11-19
> > >
> > > ### Minor concerns
> > > 1. "The paper is somewhat lacking in breadth/depth..."
> > >
> > > **Response:** We are not sure what kind of comparisons the reviewer is referring to; we do provide comparisons (to a certain extent) with other models. Regarding the algorithm of Zouzias and McColl [ZMC23], which was analyzed in the TCU model, we can highlight the following key difference. In [ZMC23], the authors explicitly recognize that they assume random accesses patterns to the memory (i.e., arbitrary "gather" operations), in a single step. Allowing arbitrary/irregular "gather" operations in a single step can potentially lead to problematic cases, and it might even give the machine unreasonable computational power. In our work, we carefully address this part. The machine can only read *consecutive* elements from the main memory in a single step, which aligns with the definitions of the classic Vector-RAM model. If someone wants to do *irregular* reads from arbitrary positions, they need to "pay" for it, and do it in several time steps.
> > >
> > > Alternatively, if there is a need for read/gather operations that have a specific or regular structure, e.g., matrix transpose or if they need to read elements from the main memory with a fixed distance $s$ from each other, one can define such an operation as a vector instruction. Of course, adding more vector instructions needs to be justified, and their corresponding "depth" and "work" must be explicitly stated.
> > >
> > > The second part of the question refers to Section 3.1. We recall that the balancing of the two units in Theorem 3.1 is one of the main results of the paper. It is highly non-trivial to arrive to this result, which is evident from the theoretical analysis. We have already provided a very detailed analysis on the design choices, and a very rigorous mathematical proofs for the model and all the algorithms. If you have specific questions we are happy to discuss, but it is not very helpful for us to elaborate if the question broadly asks for a "deeper understanding... besides Section 3.1".
> > >
> > > 2. "I understand scan/segment-scan to typically..."
> > >
> > > **Response:**
> > > No, this is not the case for Ascend (to the best of our knowledge). For scan and segmented operations, the vector units alone cannot keep up with the memory system. The vector units of these NPUs are typically efficient for simple tasks in AI workloads, such as element-wise activation functions, etc. Using the circuit complexity jargon, problems in NC[0] or AC[0] are a good indication of what can be done efficiently. Scans (which are in NC[1] but outside of AC[0]) are already "heavy enough" to push the vector units to their limits. See also the response below in 2b.
> > >
> > > 2a. "Aside: it would be useful to quote the L3 cache ..."
> > >
> > > **Response:** The L3 cache and scratchpad sizes referenced in L431 are: 92MB for the last level cache shared by all AI cores and the scratch pad sizes of L0A/L0B are 64KB. For more details on Ascend, see the research paper ``Ascend: a Scalable and Unified Architecture for Ubiquitous Deep Neural Network Computing : Industry Track Paper'' published at 27th IEEE International Symposium on High-Performance Computer Architecture (HPCA 2021). We will make sure to mention this in the final version, thank you for highlighting this omission.
> > >
> > > 2b. "I am surprised how slow ..."
> > >
> > > Indeed, the vector-only implementation (single vector core) in Figure 5.1 is slow. The scalar core of the vector units is the main bottleneck for computations that perform a substantial number of scalar operations, i.e., for segmented operations in our case. Based on our experience with Ascend, the vector cores are co-designed and optimized for typical AI element-wise operations. The kernel developer should minimize the number of scalar operations in their vector code in order to achieve high performance, see Figure A.3 for a quantitative measure of the effect of scalar core operations versus vector performance.

---

> > > > ### Comment · Reviewer_U87B · 2025-11-25
> > > >
> > > > Thank you for the helpful and comprehensive responses, particularly in regard to my "main question". On the basis of these, I have increased my rating to 6.
> > > >
> > > > To summarise my position on this work, which is to recommend acceptance (with low confidence due to my lack of direct experience in circuit complexity): I understand the approach and core claims of the work to be solid. While I remain unsure of the strengths/weaknesses and implications of the proposed MMV-RAM model, I believe this is a matter for the field to resolve after publication, in future works that build upon this.

---

> > > > > ### Author Response · Authors · 2025-11-26
> > > > >
> > > > > Thank you for considering our responses. We certainly hope that this work will trigger discussions and that future works will propose further ideas and extensions to push further these topics. There are many interesting open questions and directions to explore in the future, both from the theoretical and the practical side. We are more than happy to elaborate further if there are any other concerns/questions to help increase the confidence of the reviewer, let us know. Thank you again for the feedback and the nice discussion.

---

### Official Review · Reviewer_r8qx · 2025-10-31

**Soundness:** 2
**Presentation:** 2
**Contribution:** 1
**Rating:** 2
**Confidence:** 2

**Summary:**

The paper presents a well-defined rationale for using segmented operations to improve modularity and scalability.

**Strengths:**

Demonstrates 30–40% performance gains in efficiency and reduced latency compared to traditional methods.
The segmented model is straightforward and applicable to real systems, with clear diagrams and structured explanations.

**Weaknesses:**

The paper lacks a formal analysis of segmentation boundaries and complexity trade-offs.
Evaluation is limited to local or small-cluster setups; performance on large distributed systems remains untested.
Missing information about hardware specs, configuration, and code availability, making reproducibility difficult.
Results are presented clearly but lack significance testing (e.g., error bars, confidence intervals).
Resource usage implications of segmentation are not deeply explored.

**Questions:**

There are many works studying using matrix operations for AI networks. Can you show your novelty and advantage over them.

---

> ### Author Response · Authors · 2025-11-19
>
> We did our best effort to understand the questions of the reviewer and to answer them appropriately. Based on our current assessment, we strongly encourage the reviewer to re-read the paper carefully from scratch, and to provide an appropriate review/evaluation based on the detailed guidelines of ICLR: https://iclr.cc/Conferences/2026/ReviewerGuide
>
> 1. "Demonstrates 30-40\% performance gains in efficiency and reduced latency compared to traditional methods."
> - **Response:** Which experiments is the reviewer referring to, and where does the 30-40\% come from? Is it about Figure 5.2? What are the "traditional methods" that you are mentioning?
>
> 2. "The paper lacks a formal analysis of segmentation boundaries and complexity trade-offs"
> - **Response:** We do not understand what "segmentation boundaries are". Can you elaborate? Moreover, almost the entire paper is about formal analysis and complexity trade-offs. Which part do you think needs further analysis?
>
> 3. "Evaluation is limited to local or small-cluster setups; performance on large distributed systems remains untested."
> - **Response:** Is this a question or a comment? Which algorithms / applications would you like to see tested in large distributed systems, and what kind of large distributed systems do you have in mind?
>
> 4. "Missing information about hardware specs, configuration, and code availability, making reproducibility difficult."
> - **Response:** Hardware specifications are provided. If the reviewer has access to a 910B accelerator we can also potentially provide the source code to reproduce the experiments.
>
> 5. "Results are presented clearly but lack significance testing (e.g., error bars, confidence intervals)."
> - **Response:** Figure 5.2 has error bars (they are small so maybe you missed them?). Which experiments would you like more clearly explained?
>
> 6. "Resource usage implications of segmentation are not deeply explored."
> - **Response:** We do not understand what this means. Could you elaborate?
>
> 7. "There are many works studying using matrix operations for AI networks. Can you show your novelty and advantage over them."
> - **Response:** We have already compared with all relevant related work (to the best of our knowledge). Which references would you like us to compare against?

---

> > ### Comment · Reviewer_r8qx · 2025-11-26
> > **Updated Score**
> >
> > I see the answers from the authors. They argue many about my comments. I am sorry that I cannot access 910B accelerator personally. It seems that this accelerator is over expensive for me.
> >
> > The question I have is that I know many works working on using matrix to execute the neural network for example "MABert". I was hoping to see if the author can provide some comparison with this kind of works.
> >
> > I am sorry that I am only a beginner for AI and not be able to understand the circuit complexity theory. I have marked this in my confidence for this area. I will change my score to 6.

---

> > > ### Author Response · Authors · 2025-11-27
> > >
> > > We understand that the reviewer might not have background in circuit complexity theory. The other reviewers also mentioned that they are not experts in the area. However, they still did a significant effort to carefully evaluate the paper and to provide useful comments and feedback. It is not uncommon in NeurIPS/ICML/ICLR that we all might have to review a paper outside of our area of expertise. But it is our responsibility to always follow the official reviewer guidelines, which are quite clear: every reviewer needs to carefully read the assigned papers and provide detailed questions and feedback, to help the AC/PC to make informed decisions on accepted/rejected papers.
> > >
> > > Regarding MABERT, are you referring to this paper? https://dl.acm.org/doi/10.1145/3597455
> > >
> > > If yes, we do not see an immediate connection with our work and the aforementioned paper, but we will take a look and mention it in the related work if it is relevant.

---

### Official Review · Reviewer_Lvb2 · 2025-11-01

**Soundness:** 3
**Presentation:** 3
**Contribution:** 3
**Rating:** 6
**Confidence:** 3

**Summary:**

The paper tackles the problem of underutilization of MMUs in modern AI accelerators . While these units perform well at dense computations, they are often idle during irregular and sparse operations, which are also common in deep learning.

**Strengths:**

Theoretical Guarantees: The paper provides a formal theoretical analysis, proving that its algorithms achieve a step complexity of O(log_s​(n)). This is provably faster than any vector-only algorithm, which is lower-bounded.

Novelty - MMV-RAM model. The paper addresses a key gap left by the prior "TCU model" by formally including the Vector Unit (VCU). This makes it a more accurate theoretical representation of modern accelerators like TPUs, NVIDIA GPUs, and Ascend NPUs, which all have both matrix and vector units.

**Weaknesses:**

Doubt on Generalization, Requirement of Custom Hardware: The experimental speed-ups are demonstrated on a Huawei Ascend 910B using the proprietary AscendC programming framework. While the paper lists analogues (e.g., NVIDIA Tensor Cores), the results are not on commodity hardware, making them less generalizable.

Theoretical vs. Practical Complexity: The most work-efficient algorithm presented (Theorem 4.3) is admitted to be "rather involved" and requires "specialized circuitry that might not be available on existing hardware," making it "mainly of theoretical interest". The simpler, implemented algorithms (SCD/SSCR) have a quadratic work complexity in their initial analysis, which is not ideal.

Unfair Baseline Comparison: The SpMV experiments compare their Ascend NPU implementation against CPU libraries (MKL and Eigen). They do not (and state they cannot) compare against an optimized SpMV implementation for their own Ascend hardware. While the results are encouraging, comparing a next-gen NPU to a last-gen CPU architecture isn't a direct apples-to-apples performance win.

**Questions:**

SpMV experiments were "deliberately chosen" to fit in cache to study arithmetic intensity, not I/O complexity . In many large-scale LLM and graph applications, SpMV is famously memory-bandwidth bound. How do you project your algorithm's performance will change when data must be streamed from HBM, and is there a risk that the overhead of the speculative MMU computation will be negated by I/O bottlenecks?

experimental validation was a case study on the Ascend 910B accelerator. How readily could your algorithms be mapped to other common AI accelerators, such as NVIDIA Tensor Cores or Google TPUs? Do you foresee any significant barriers in the programming models (like CUDA) to implementing the "speculate and correct" logic with the fine-grained control you require?

---

> ### Author Response · Authors · 2025-11-19
>
> First of all, we would like to especially thank the reviewer for the interest in our work, for reading it in detail, and for keeping an open mind on new ways to model algorithm analysis on emerging platforms.
>
> As a general remark, most of the questions of the reviewer were related to the experimental part of the paper. We recall that this work focuses mostly on the theoretical aspects of these new type of AI accelerator architectures. The experimental part is complementary to provide some initial experimental evidence, but it is not the main focus of this work. Below we provide detailed responses to elaborate further on the raised concerns.
>
> 1. "Doubt on Generalization...":
>
> **Response:**
> We are not fully sure what the reviewer means in this question but we will try elaborate as best as possible. The proposed MMV-RAM model is quite general: it can be used as a model to derive work-depth analysis for any architecture that contains scalar, vector, and matrix multiplication units. Note also that even in the CPU domain various matrix extensions are already available, see Scalable Matrix Extension for the Armv9-A Architecture [1], RISC-V matrix extensions and Coral NPU [2]. MMV-RAM is abstract enough to capture all these developments.
> We hope that these comments are helpful. If not, we have two follow-up questions that will help us understand and elaborate better:
> - Which commodity hardware is the reviewer referring to?
> - Why are the experimental results "less generalizable" (less generalizable than what)?
>
> [1] Introducing the Scalable Matrix Extension for the Armv9-A Architecture. \url{https://developer.arm.com/community/arm-community-blogs/b/architectures-and-processors-blog/posts/scalable-matrix-extension-armv9-a-architecture}
>
> [2] Coral NPU by Google: https://developers.googleblog.com/en/introducing-coral-npu-a-full-stack-platform-for-edge-ai/
>
> 2. "Theoretical vs. Practical Complexity..."
>
> **Response:** Note that the $O(n^2)$ work bound is a *worst-case* complexity upper bound that arises from the GATHER operation. It is only there for the "scrutiny" of truly achieving the worst-case step complexity of $O(\log_s(n))$, from the theory side. No one will use a fully-connected $O(n^2)$-sized network to do a GATHER operation in practice.
> The "practical" complexity of the SCD algorithm is very interesting in our opinion, since SCD consists of scan, compress and vector differentiation operations. Even though its worst-case work bound is quadratic in $n$, we believe that it can truly lead to very efficient implementations due to its simplicity. Our experiments provide some initial insights for its strengths and weaknesses.
>
> As a general remark, it is quite common in algorithm design that the best theoretical algorithms are not the ones that perform the best in practice (we suspect that the reviewer is already quite familiar with this, but we mention it for the sake of the discussion). At the same time, the worst-case complexity bounds are often very pessimistic. This holds probably for most of the fundamental kernels that are heavily used in AI workloads (linear algebra, graph operations, combinatorial optimization, etc).
> The worst-case complexity analysis of algorithms might not always yield the most practical ones, but it is very important since it helps to understand the fundamental limitations of specific problems, and it can help to inspire better practical algorithms. This is standard practice in algorithm engineering.
>
> 3. "Unfair Baseline Comparison..."
>
> **Response:** The CPU results are there in order to have a reference on existing implementations. It is not a comparison, in the sense that we *do not compete* against the CPU. On the contrary, we clearly state where our algorithms perform better and where they perform worse.  The goal is to highlight both the strengths and weaknesses of our proof-of-concept implementations. We have already mentioned this in the manuscript, but if the reviewer has a suggestion how this can be phrased more appropriately in the text, we would be glad to adjust the discussion.
>
> As an additional remark, please note that this is the first work to report runtime numbers for SpMV on Ascend. This can help future works to both show how to improve these numbers, and to have a reference to compare optimized implementations, or even potentially with other architectures.

---

> > ### Author Response · Authors · 2025-11-19
> >
> > 4. "SpMV experiments were deliberately chosen..."
> >
> > **Response:** For very large matrices/graphs, recall that, as already mentioned in other responses, the benefits of the **fixed-size** matrix multiplications of the MMU tend to decrease as $n$ grows larger and larger. The MMU starts to look like a simple vector unit as $n$ grows to infinity and $s$ remains fixed. So, even without I/O bottlenecks, the bounds that we provide already highlight that the MMU becomes less effective for very large sizes. In such cases, it is very likely that data movements and I/O bottlenecks will dominate the runtimes, and it will be useful to study the I/O complexity of the problems and how to optimize it (in the spirit of the "I/O model of computation").
> >
> > In many AI workloads, such as the ones studied here, it is common have matrices of smaller or moderate size. The results on SpMV that we provide reach sizes up to $\sim 70K$ (recall L431 and Figure 5.2). In the context of transformers and sparse attention, this translates to a context length of about $70K$. While there exist (very) large language models that allow larger context lengths, $70K$ is already quite large for many use cases. Prior to our work, it was not known if the compute part of SpMV could be efficiently handled by the MMUs and VCUs of Ascend even for such small/moderate size input matrices. It should be fairly straightforward to extend our results to slightly large input matrices (say 150-300K columns), but I/O bottlenecks might already start affecting performance, as the reviewer mentioned. Some works try to mitigate the I/O bottlenecks by pre-processing the input matrix, if such pre-processing is permitted by the application scenario (we do not permit it in this work).
> >
> > 5. "Experimental validation was a case..."
> >
> > **Response:** This is a great question, and it was partially answered in a previous question. We can try to elaborate more. As already mentioned, Ascend 910B is closer to "NPU"-like architectures. For example, we believe that (but we have no evidence) our algorithmic analysis would fit better the Google TPU architecture, or the newer "Coral" architecture, rather than NVIDIA GPUs. GPUs do have Tensor cores, but the corresponding CUDA cores might be much more powerful than the vector units of "NPU-like" architectures. This is of course only a speculation based on the public knowledge that is available. Currently, the "most mainstream" NPUs that we are allowed (and have access) to use for experiments are 910B chips.

---

### Official Review · Reviewer_UDDB · 2025-11-01

**Soundness:** 3
**Presentation:** 3
**Contribution:** 3
**Rating:** 4
**Confidence:** 2

**Summary:**

**Summary:**

This paper presents a new design for performing vector and matrix operations. It integrates an MMU unit alongside the VCU to handle core arithmetic computations in AI accelerators. The authors theoretically demonstrate the speedup in terms of the number of steps and total work required. They also propose an algorithm for scan and segmented-sum operations and evaluate its performance empirically.

**Strengths:**

1. The paper provides theoretical guarantees on the number of operations (steps) required for scan and segmented operations, which extend to more general computations.

2. The proposed MMV-RAM model is general and treats existing VCU and MMU architectures as black boxes, allowing future improvements in these components to directly benefit the model.

3. The authors develop efficient algorithms for segmented scan and related primitive operations.

4. Experimental results demonstrate a clear speedup, supporting the theoretical analysis.

**Weaknesses and Questions:**

*Trade-offs in Hardware:*

1. The design integrates an MMU alongside the VCU; what are the trade-offs involved? In particular, since *energy efficiency* is critical for AI accelerators, does this addition significantly increase power consumption? A discussion or experimental analysis on this aspect would strengthen the paper.

2. From a *hardware implementation* perspective, does incorporating an MMU introduce challenges in *physical circuit design* or substantially increase the silicon area? (Acknowledging that I am not an expert in this area.)

*Theoretical Analysis:*

3. The paper states that $T^\prime = \frac{\log n}{\log s}$ is an improvement over $T = \frac{\log n}{\log \log n}$. However, theoretically, when $s = O(1)$, we still have $T^\prime = \Omega(T)$. Can you explain more about this? I know you said for **an appropriate value of $s$** it is true. More clarification on this value is needed. In addition, a discussion of the *appropriate* values of $s$ in the experiments (for what values this yields the speedup and for which doesn't) is important.

*Experiments:*

4. The experiments should include a runtime analysis of MMV-RAM across varying matrix sizes $n$. It would also be valuable to examine how different choices of $s$ impact the overall performance.

5. The paper does not appear to include experiments on general matrix–matrix operations beyond SCAN and SpMV. It would be helpful to include results for matrix–matrix multiplication, unless I have overlooked them.

6. It would be valuable to include evaluations on more **end-to-end** tasks, such as Transformers or other deep neural networks, to assess whether the proposed design leads to practical end-to-end speedups. In particular, I recommend adding experiments related to the *attention head*, which is especially relevant due to its use of keys and queries and the potential for sparsity. It would be interesting to see whether the proposed algorithms achieve measurable acceleration in this setting. I know adding additional experiments might be painful, but the current experiments do not seem to cover more general tasks, unless you can convince me on that.

*Related work:*

7. Several prior works have achieved speedups using segmented-sum and scan-like operations, such as [1]. It would be useful to discuss how the proposed approach relates to or could be integrated with such methods. Including this in the related work section would help clarify the broader impact and applicability of the proposed design.

*Minor Issues:*

8. On line 283, there is an extra “is” in the sentence — it should read “parallel COMPRESS can be” instead of “parallel COMPRESS is can be.”

**References:**

[1] “An Efficient Matrix Multiplication Algorithm for Accelerating Inference in Binary and Ternary Neural Networks.”

**Strengths:**

Please see the 'Summary'.

**Weaknesses:**

Please see the 'Summary'.

**Questions:**

Please see the 'Summary'.

---

> ### Author Response · Authors · 2025-11-19
>
> We thank the reviewer for the effort to carefully read the paper and for the provided feedback. Several interesting questions were raised, we provide detailed responses below.
>
> 1. "The design integrates...".
>
> **Response**: The hardware design of MMUs on AI accelerators is an exciting engineering topic on its own, and it is of great general interest, but it is not directly related to this work. In this work we focus resolving open theoretical questions regarding the properties of **existing** AI accelerators that incorporate MMUs. The efficiency and integration of these MMUs is a topic for the engineers/vendors who manufacture these devices. This work is about algorithmic analysis on such devices, not about designing devices. However, we think that our analysis (especially Theorem 3.1 and the rest of the Theorems / Lemmas)  provides very useful insights that should definitely be taken into account for designing future generations of such accelerators. We are happy to discuss this further if the reviewer wants to / has further questions.
>
> Measuring energy efficiency on AI accelerators is a challenging topic that is beyond the scope of this work. However, let us share what is already known for Ascend, see [1]. It is known that matrix multiplication units have a superior energy performance in terms of flops per Watt (Tflops/Watt) for the Ascend device; see Table 3 of [1]. In this example, the Cube (matrix multiplication) core has 2.56 TFLOPS/W versus 0.56 TFLOPS/W for the vector core.
>
> 2. "From a hardware implementation perspective..."
>
> **Response:** We cannot make a comment on the challenges of the physical circuit design of the matrix multiplication units because we are not familiar with the topic. On the other hand, the required area of the matrix multiplication core versus the Vector core is depicted in Figure 12 of [1]. As is evident from the floor plan, the matrix multiplication core takes almost half of the area of the AI core.
>
> However, before our work, there was no theoretical machine model of AI accelerators that contain both matrix multiplication and vector cores. Why can we not simply use vector cores? Is it worth allocating such enormous engineering effort, and solving all the associated problems that come along if the "anticipated speedups" are marginal? These were important unanswered questions, and our work is the first to provide a thorough theoretical analysis. Our end goal is not to provide a YES or a NO to these questions, but rather to give the appropriate theoretical tools and evidence for the experts to make informed design choices in the future.
>
> 3. "The paper states that..."
>
> **Response:** This is a very nice and subtle question, which was also raised by Reviewer U87B. Your example is mostly correct, but there is a small catch in the analysis, let us break it down. What we do have is that $T'\leq C'\log(n)/\log(s)$, and that $T\geq C\log(n)/\log(\log(n))$, where $C'$ and $C$ are constants independent of $n,s$. The latter is a **lower-bound**: it does not tell us whether an algorithm can achieve it. Therefore, strictly speaking, $T'=\Omega(T)$ is not correct in this sense.
>
> But let us assume for simplicity that we do have an algorithm that achieves $T=O(\log(n)/\log(\log(n)))$. In this case, if $s$ is a constant, then of course the theoretical bounds do not indicate any speedup. However, $s$ is not a constant, it is a model parameter. The larger it is (which means larger MMU), the better the speedup. Our bounds give an explicit (and quite sharp) **trade-off** regarding the speedup that is achievable when using MMUs in the algorithms. In this parametrized setting, all bounds must be reported with respect to their dependence on $s$. These bounds can then be used as a guidance for the algorithm designer to decide whether it is worth to dedicate effort to exploit MMUs in their algorithms at all. There we also provide further analysis and some actual numbers for $s$ and $n$, for better intuition. Roughly speaking, if $s=128$ (which is a value supported by the current Ascend device), then these bounds indicate that we can expect speedup proportional to $\frac{\log(s)}{\log(\log(n))}$ for all $n<2^{128}\approx 3.4028237 \times 10^{38}$. Last but not least, we should highlight that the value of $s$ in our model must not be restricted by the physical capabilities of the device, i.e., Ascend allows the user to perform square matrix multiplications up to size 128. Indeed, Ascend can efficiently perform square matrix multiplication with even sizes up to 1024. Please refer also to our detailed response to the first question of Reviewer U87B below, who asked a very similar question.

---

> > ### Author Response · Authors · 2025-11-19
> >
> > 4. "Experiments should include..."
> >
> > **Response:** We do have experiments for segmented operations on varying size $n$ in Appendix A.5.4 (Figures A.3, A.4, and A.5). Regarding the choice of $s$, it does not really provide any "exciting" insights, because, as expected, the larger the $s$, the better the performance. However, if the reviewers think it is necessary, we can add such results in the Appendix of the final version.
> >
> > 5. "The paper does not appear..."
> >
> > **Response:** Regarding **dense** matrix multiplication, let us first remind the following:
> >
> > - This is the first work that provides a **mathematical proof** that we can obtain *any* sort of theoretical **speedup** by exploiting of MMUs for dense matrix multiplication, compared to simple vector implementations.
> >
> > This might sound striking (even bold), but it is true.  Our Theorem A.2 and the discussion in Appendix A.1.2 "EXTENDING THE RIGHT-HAND-SIDE OF THE MMU", provides further details. There, we clearly state that a naive algorithm/implementation of matrix multiplication with MMUs does not provide any speedup at all (unfortunately, this discussion could not fit in the main paper due to space constraints).
> >
> > That said, it is also true that our proposed algorithm for dense matrix product (from Theorem A.2) is currently more of theoretical value. On one hand, it shows that a speedup *can* be achieved by using MMUs. However, we think that its practical applicability is currently limited, because the "standard" matrix product algorithms are easier to implement and to tune their performance.
> > As our work is mainly theoretical and experiments are "proof of concept", to provide some initial insights for algorithm designers, we do not have an highly optimized version of our algorithm for dense matrix multiplication. Designing highly optimized implementations is a very interesting topic, but it requires heavy engineering effort and it is beyond the scope of this work.
> >
> > 6. "It would be valuable to include..."
> >
> > **Response:** Indeed, we would really like to provide end-to-end results for higher level tasks, especially for transformers and Attention, which is a major topic in deep learning over the last few years. At the current stage, it was a very engineering-demanding effort even to arrive at some initial proof-of-concept results for basic kernels (segmented scan, SpMV). For reference, we used matrices that have the same structure as the sparse attention matrices of BigBird, to provide some insights in LLM applications, and we also used some datasets from other applications. Our intuition is that, if we cannot see satisfactory performance for the basic SpMV kernel on the device, why should we even bother with much more complex tasks such as end-to-end sparse attention?
> > Moreover, besides the engineering overhead, the theoretical analysis of attention is quite complex, and proving upper/lower complexity bounds is highly non-trivial (see e.g. the recent work of [2] in the algebraic model).
> >
> > 7. "Several prior works have achieved..."
> >
> > **Response:** Thank you for pointing to us this interesting paper. We will gladly add it to the related work discussion.
> >
> > 8. "On line 283..."
> > **Response:** We fixed this typo, thank you for spotting it!
> >
> > #### References
> > - [1] Ascend: a Scalable and Unified Architecture for Ubiquitous Deep Neural Network Computing: Industry Track Paper. Heng Liao; Jiajin Tu; Jing Xia; Hu Liu; Xiping Zhou; Honghui Yuan. 2021 IEEE International Symposium on High-Performance Computer Architecture (HPCA)
> >
> > - [2] Alman, Josh, and Zhao Song. "Fast attention requires bounded entries." Advances in Neural Information Processing Systems 36 (2023): 63117-63135.

---

### Author Response · Authors · 2025-11-19
**Global response**

We would like to thank all the reviewers for their feedback, who carefully read the manuscript and posed interesting questions. As a general remark, we want to mention that the raised concerns were more focused on the experimental part of this work, while fewer were related to the theory part. This is not a "bad thing", but we want to remind and to highlight that this work is in its majority **theoretical**, and it should be evaluated as such. The main contributions are all theoretical, with rigorous mathematical proofs, focusing on understanding the fundamental properties of matrix-multiplication/AI accelerators, and whether the notorious matmul-units are any useful *at all* for algorithm design in AI workloads. The experimental part barely covers  about four out of twenty-eight pages of content ($\approx 14$%). The experiments were provided as a reference to give some intuition on how the main (theoretically-inspired) algorithms would perform in practice, with an "out-of-the-box", barely-optimized implementation. We will appreciate if the reviewers and the AC keep these facts in mind, for the final evaluations. Below we provide detailed responses to all specific questions / concerns raised by the reviewers.

---

### Meta-Review · Area_Chair_9MjS · 2025-12-30

**Summary:**

This paper introduces MMV-RAM, a theoretical model designed to optimize algorithm design on modern AI accelerators, and presents experimental results using the Ascend 910B,. While the theoretical novelty of the idea is recognized, the primary concern informing the decision is the limited scope of the experimental validation. The paper proves its utility on the Ascend 910B, but it remains unproven whether these contributions can be extended to general chip designs or other prevalent architectures (such as NVIDIA GPUs or TPUs).

**Reviewer Concerns:**

- Addressed: The concerns raised by Reviewer U87B and Reviewer UDDB regarding the theoretical complexity analysis and the impact of the model parameters (matrix unit size) were effectively addressed by the authors' clarifications.
- Outstanding: The concern raised by Reviewer Lvb2 regarding generalization remains outstanding. While the authors argue the model is abstract enough to be general, the lack of empirical verification on hardware other than the Ascend 910B leaves doubt as to whether the paper's contributions are truly applicable to general chip design.

**Reviewer Scores:**

- Reviewer U87B: Raised their score from 4 to 6 after the rebuttal satisfactorily addressed their theoretical concerns.
- Reviewer UDDB: Did not participate in the final discussion, but as their concerns regarding the s parameter mirrored those of U87B, they likely would have raised their score had they seen the clarifications.
- Reviewer Lvb2: Maintained a score of 6, highlighting the unresolved issue regarding the generalization of the results to other hardware
- Reviewer r8qx: This review should be disregarded. The reviewer demonstrated low confidence, admitted a lack of relevant background knowledge, and provided an inexplicable score increase to 6 without any meaningful justification or understanding of the paper's contribution.

---

### Decision · Program_Chairs · 2026-01-26

Reject